# Genotype-fitness mapping of adaptive mutants reveals shifting low-dimensional structure across divergent environments

**Olivia M. Ghosh**[1,2,3*], **Grant Kinsler**[2,4], **Benjamin H. Good**[2,3,5], **Dmitri A. Petrov**[2,5*]

1 Department of Physics, Stanford University, Stanford, California, United States of America,
2 Department of Biology, Stanford University, Stanford, California, United States of America,
3 Department of Applied Physics, Stanford University, Stanford, California, United States of America,
4 Department of Bioengineering, University of Pennsylvania, Philadelphia, Pennsylvania, United States of America, 5 Biohub, San Francisco, California, United States of America

* omghosh@stanford.edu (OMG); dpetrov@stanford.edu (DAP)

## Abstract

A central goal in evolutionary biology is to predict the effect of a genetic mutation on fitness. This is a major challenge because it requires knowledge of both the phenotypic effects of a mutation and their importance in an arbitrary environment, which are high-dimensional quantities and difficult to guess *a priori*. Here, we address this problem by taking a top-down, data-driven approach to infer the mapping between genotypes, latent phenotypes, and fitness. We measure the fitness effects of a large collection of adaptive yeast mutants in many lab environments, from which we build low-dimensional, linear fitness landscapes. We find that these models are highly predictive of fitness variation for thousands of adaptive mutants, both in environments similar to where they evolved and also in divergent environments. This implies that the underlying genotype-phenotype-fitness maps for these adaptive mutants tend to be broadly low-dimensional. We further demonstrate that these maps only partially overlap across divergent environments, suggesting that the phenotypic determinants of fitness shift with the environment but remain low-dimensional. These results combine to emphasize the importance of environmental context in evolution, and suggest that top-down, low-dimensional fitness landscapes pave the way for evolutionary prediction.

## 1 Introduction

A fundamental challenge in evolutionary biology is developing a predictive framework that links genetic variation to phenotypic traits and, ultimately, to evolutionary fitness [1–7]. One major obstacle is the sheer size of genotype space [8–10]. Each genetic variant can modify molecular phenotypes, which in turn affect higher-level phenotypes, eventually causing macroscopic changes to organisms. Building bottom-up, causal chains across these phenotypic levels is extremely difficult, given the vast network of interactions at every level of biological organization. Of particular interest in evolutionary biology is understanding how genetic changes affect phenotypes that matter to fitness [11,12]. An additional layer of complexity enters

**Data availability statement:** Data underlying all figures can be found in 10.5281/zenodo.18175930. All other metadata, as well as the source code for the fitness inference pipeline, downstream analyses, and figure generation, are available at Zenodo (10.5281/zenodo.18175930) or GitHub (https://github.com/omghosh/limiting-functions). The software repository for the barcode counting code can be found at Venkataram et al. (2020). (https://github.com/sandeepvenkataram/BarcodeCounter2).

**Funding:** This work was supported by the National Institutes of Health NIGMS (https://www.nigms.nih.gov/) grant no. 5R35GM118165-07 to D.A.P., National Institutes of Health NIGMS grant no. R35GM146949 to B.H.G., and National Science Foundation Graduate Research Fellowship Program (https://www.nsfgrfp.org/), grant no. 2020301250 to O.M.G., B.H.G., and D.A.P. are Biohub - San Francisco Investigators (https://biohub.org/). The funders had no role in study design, data collection and analysis, decision to publish, or preparation of the manuscript.

**Competing interests:** The authors have declared that no competing interests exist.

**Abbreviations:** FDR, false discovery rate; PCA, principal component analysis; SSE, squared sum of errors; SVD, singular value decomposition; TSS, total sum of squares; WT, wild type.

with the environment, which can modulate how genotypes map onto phenotypes, and their mapping onto fitness [13–15]. Hence, a genotype-phenotype-fitness map must contend with environmental and ecological variables, which are numerous, interdependent, and dynamically shifting. Through this lens, the goal of building a predictive model of evolution appears impossible, stifled by biology's curse of dimensionality.

Despite this challenge, work from recent decades has often uncovered low-dimensionality in biological systems [16,17]. From the molecular level with protein structures and configurations [18–20], to simple physiological growth laws [21] and modular gene expression [22], to canalization in development [23,24], all the way up to the ecological community level [25], low-dimensional structure has emerged across scales. Because these systems are themselves produced and shaped by evolution, it is plausible that evolutionary processes are effectively governed by a small number of latent variables [26,27]. Recent empirical work supports this view, finding low-dimensional structure in genotype–phenotype and phenotype–fitness maps across diverse systems [28–35]. While the underlying mapping between genotype, phenotype, and fitness is likely to be nonlinear [36], many of these studies have successfully modeled the fitness of a particular genotype as a linear function of $D$ underlying, latent phenotypes,

$$X(\vec{g}, \vec{E}) = \sum_{k=1}^{D} \phi_k(\vec{g})\beta_k(\vec{E}),$$

(1)

where $\phi_k(\vec{g})$ is genotype $\vec{g}$'s effect on latent phenotype $k$, $\beta_k(\vec{E})$ is the weight of that latent phenotype in environment $\vec{E}$, and $D$ is the total effective dimensionality of the latent phenotype space.

These linear genotype-phenotype-fitness maps can be inferred via matrix factorization from large datasets composed of the fitness of many genotypes in many environments. This approach circumvents the difficulty of defining which phenotypes *should* be measured, and instead infers the underlying phenotypic structure from fitness variation. One study from our lab looked at a collection of experimentally-evolved adaptive yeast mutants and found that subtle environmental perturbations near the original evolution condition (Fig 1A) were enough to induce variation in fitness that contained signals of the underlying latent phenotype space [28]. We referred to these latent phenotypes, $\vec{\phi}$, as "fitnotypes," because they are fundamentally different from traditional, measurable phenotypes. Indeed, fitnotypes are only detectable as latent variables that contribute to fitness variation, while phenotypic changes that do not affect fitness are hidden. These fitnotypes, in turn, determine fitness based on environment-specific weightings (see Supporting information for an explication of this top-down approach). If adaptive mutants are functionally identical because they evolved towards the same fitness optimum, they should exhibit congruent behavior in fitness space regardless of the number of environments tested, and hence a one-dimensional version of Eq. (1) would suffice to capture fitness variation across multiple environments. Other studies have used a similar model to Eq. (1) to

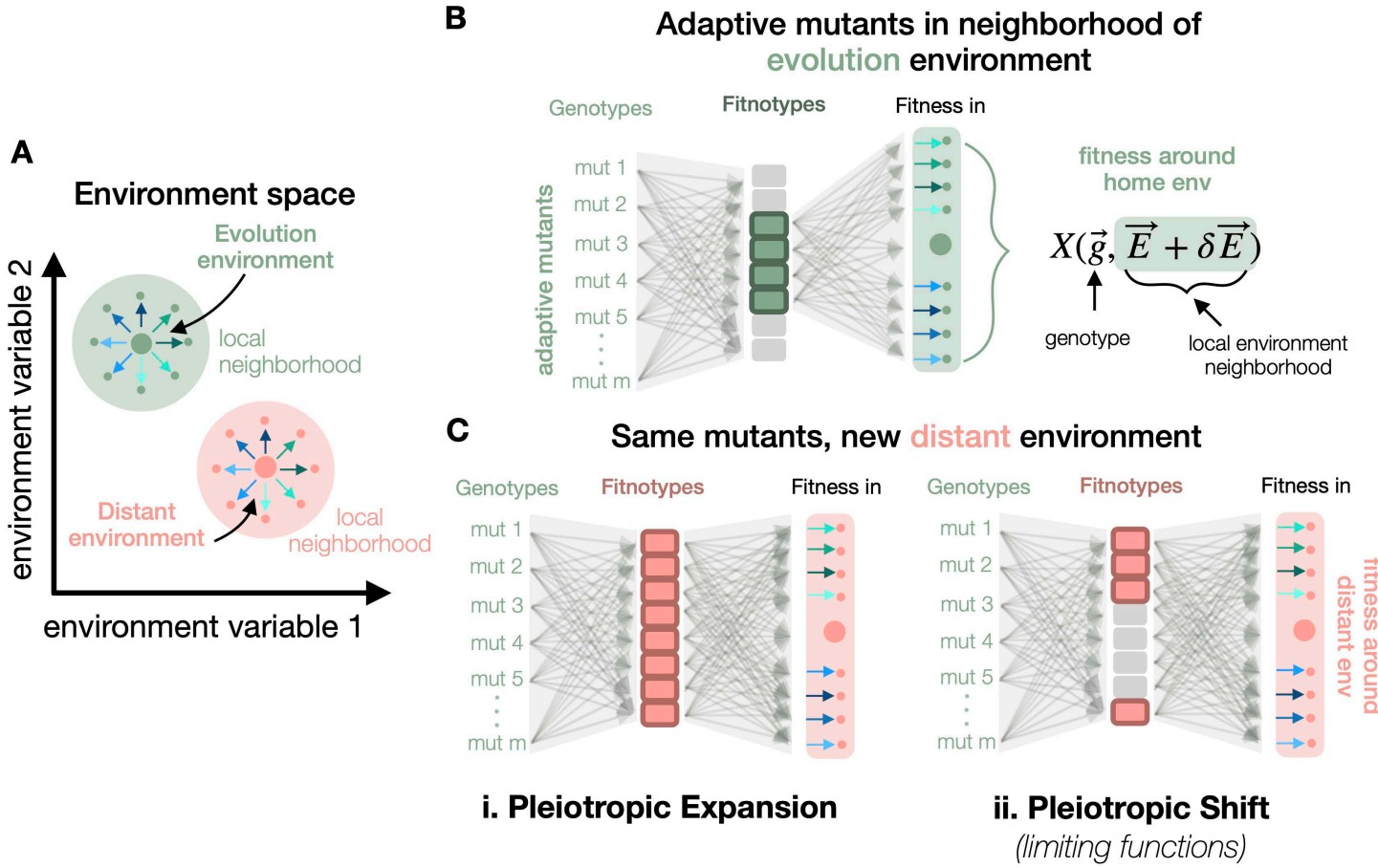

**Fig 1. Two models for the nature of pleiotropy in adaptation.** (A) Schematic of the environmental structure in this study. Environments can be mapped onto a multidimensional environment space characterized by chemical and physical compositions. The large green circle represents an environment where adaptive mutants evolved, and the large pink circle is a distant environment. Around each base, a set of identical environmental perturbations (arrows) is applied, generating clusters of similar environments around distinct base environments. (B) Schematic of fitotype map for adaptive mutants near their home base environment. By measuring fitness in each of the green environments, we can infer how many fitotypes matter for this set of mutants in their home environment. Here, only four of the possible 8 fitotypes matter. (C) When we move the mutants to the distant base environment, and measure their fitness in all pink environments (base and perturbations), there are two possibilities. Either more fitotypes become important and the space appears higher-dimensional (left, pleiotropic expansion), or the set of fitotypes that matters remains low-dimensional, but shifts (right, pleiotropic shift).

capture the fitness of thousands of QTLs from a yeast cross across 18 environments, or the behavior of human cell line knockouts under genotoxic stressors [29,30,37], often including different constraints such as sparsity in the factorization problem.

All together, these studies suggest that low-dimensional structure underlies genotype-phenotype-fitness mapping across environments. But the source of this low-dimensional structure remains unclear, particularly in the case of adaptive mutants in subtle environmental perturbations [28].

One explanation is the "pleiotropic expansion" model. In this framework, mutations selected in one environment (Fig 1A, green environments) are initially constrained to a low-dimensional phenotypic space, where only a subset of phenotypic effects matters to fitness (Fig 1B, top). However, when placed in a new base environment with the same perturbations as above (Fig 1A, pink environments), their pleiotropic effects become unconstrained, leading to an expansion in the number of phenotypes that influence fitness (Fig 1B, bottom left). If true, this suggests that evolution dynamically

constrains mutants to explore the "flattest" paths in phenotype space within a given environment, eliminating the costs associated with affecting too many phenotypes [38]. Yet, in novel environments, no such constraints exist, allowing the full phenotypic diversity (and likely the associated costs) of a mutation to be exposed. This model assumes that the space of possible phenotypes that could affect fitness, especially negatively, is very large, consistent with countless observations of widespread pleiotropy and polygenicity [11,12,39]. But it also allows for adaptation to occur via an effectively low-dimensional phenotype space in the face of this complexity, which theoretical work suggests may come at a cost to fitness [38,40]. In other words, the "pleiotropic expansion" model posits that there are limited ways to improve fitness, but many ways to decrease fitness. In this model, the source of low-dimensionality is short-term evolutionary dynamics, where ascertainment bias in phenotypes skews adaptive mutants to appear low-dimensional only in their home environment. While the reverse model, a "pleiotropic compression," is theoretically possible, previous work has suggested that for this particular set of mutants, it is unlikely [28]. Such a model would also be difficult to detect, because fitnotype space around the evolution environment is already low-dimensional, and any further compression would be minimal.

An alternative explanation is the "pleiotropic shift" model (Fig 1B, bottom right), in which adaptive mutants always affect many phenotypes, but only a small subset of these phenotypes are relevant in any given environment. As environments change, the relative importance of different phenotypes shifts, reweighting their contributions to fitness. Under this model, genotype-phenotype-fitness maps are consistently low-dimensional, both in the home environment and in distant environments, because fitness is determined by a small number of dominant, environment-specific challenges. This model bears resemblance to previous work on linear pathway models and Liebig's law of the minimum, which states that growth is limited by the first essential nutrient or metabolite to run out [41–45]. In analogy to this work on limiting nutrients or limiting metabolites, we conceptualize this pleiotropic shift model as indicative of "limiting functions." In this qualitative model, the set of functions that a cell must perform in order to thrive is large, but at any given time there will be a small number of functions that limit fitness. Thus, only a small number of limiting functions can be profitably modified in adaptation. Meanwhile, the pleiotropic side-effects on nonlimiting functions are effectively hidden from natural selection. Just as nutrients can shift from being in excess to being limited as the environment changes, so too can functions shift between being nonlimiting and limiting in new environmental contexts. Such a model has implications for evolution across multiple environmental epochs, and bears on the question: how likely is it that the important functions in one environment will be important in all environments? In this model, the source of low-dimensionality is long-term evolution, which has shaped physiology to respond to environments in a generically low-dimensional way.

To differentiate between these two scenarios, we must build distinct fitnotype spaces for at least two different "base" environments, as in Fig 1A, wherein adaptive mutants are measured in environments around their home base (evolution environment), and another distant base. By comparing different latent fitnotype spaces, we can characterize how the environment modulates genotype-phenotype-fitness mapping (Fig 1B). In this work, we used a set of adaptive *Saccharomyces cerevisiae* mutants from previous evolution experiments, with known origins. We selected the home base and two additional "distant" base environments, chosen to modify the growth cycle and salinity of the environment. To each base, we applied an identical set of environmental perturbations (changes in temperature, glucose concentration, etc). We used singular value decomposition (SVD) to infer three separate latent fitnotype spaces for each base environment from fitness variation due to the perturbations, and compared their dimensionality. We further used predictive power as a proxy for estimating the overlap in fitnotype space across these base environments. Our findings ultimately reject the pleiotropic expansion model and strongly support the pleiotropic shift model. We interpret these results as evidence in favor of a "limiting functions" model of fitness. Environments impose a low-dimensional constraint on evolution, as each environment presents a limited number of dominant challenges that shape fitness outcomes. These results highlight the fundamental role of environmental context in shaping the genotype-phenotype-fitness landscape, providing insight into how adaptive evolution navigates complex, high-dimensional spaces.

## 2 Results

### 2.1 An experimental design to probe the environmental modulation of fitnotype space

To test whether adaptive mutants undergo a pleiotropic expansion or shift when they are moved away from their home environment, we designed an experiment to infer fitnotypes (latent phenotypic dimensions) for adaptive mutants in different "base" environments. In this study, we compiled a library of roughly 4,000 adaptive, DNA barcoded *S. cerevisiae* mutants from previous evolution experiments [46–48]. These mutants evolved in the same glucose-limited media, under a serial dilution growth regime portrayed in Fig 2A, but they differ in the time between passages ($T_{transfer}$). Previous work has shown that fitnotypes can be inferred from fitness variation due to subtle perturbations around a base environment, so we chose three "base" environments around which to build our fitnotype maps: the two evolution conditions ($T_{transfer}$ = 48 hours: "2 day," and $T_{transfer}$ = 24 hours: "1 day"), along with one more extreme environment, "Salt," which has 0.5 M NaCl added to the glucose-limited media and a transfer time of 48 hours. We applied a set of roughly 20 environmental perturbations to each base environment, which comprises the columns in Fig 2B. These perturbations included small alterations to incubation temperature (30 °C ± 2 °C), in glucose concentration (1.5% ± 0.2%), or introducing sub-inhibitory amounts of drugs. See S1 Fig for more details about the environments.

To measure the relative fitness of all mutants in the library in a single environment, we utilized previously established methods that leverage DNA barcoding and high-resolution lineage tracking. We competed a pool of 4,000 barcoded strains against a highly-abundant reference strain, the wild type (WT) ancestor, which took up 95% of the culture. We quantified the frequencies of barcodes over time via amplicon sequencing. Based on these frequency trajectories, we inferred the relative fitness of thousands of strains. Fig 2A shows the workflow from serial dilution experiments to highly replicable fitness inference [28,49,50]. See Methods for more details on the fitness assay.

Fig 2B shows a representative subset of our data, where each row is a mutant and each column is an environment. The mutants are organized by their background genotype and evolution environment, or home base, and the environments are organized by base. The most obvious signal comes from the tradeoffs present in Salt conditions, where mutants that are adaptive in 1 or 2 days are much more likely to be deleterious with respect to the ancestor in the Salt base. This genotype-by-environment fitness matrix will be the basis of our further analyses.

First, we characterized each environment as an *m*-dimensional fitness vector, where *m* is the total number of mutants. This mutant-embedding of environment space is helpful because it allows us to focus solely on environmental differences that matter to fitness for our focal set of adaptive mutations. We used principal component analysis (PCA) to project each environment vector into a 2-D space, onto the first two PCs. In Fig 2C, each point is one environment (unique base-perturbation combination). We colored each environment according to its base, and found that each set of perturbations belonging to the same base remains mostly localized in this two-dimensional space. This is especially informative because the first two principal components explain nearly 82% of the variance in the data. 1 and 2 days are more similar bases to each other, as was evident from the heatmap in Fig 2B, and Salt is a more distinct base. Therefore, our set of environments provides a range of environmental scales across which we can compare fitnotype maps of adaptive mutants.

Due to the large number of environments in our assay, we performed fitness measurements in batches of different perturbations, each containing one "un-perturbed" base, lending us batch replicates (diamonds in Fig 2C). These serve as a natural baseline of variation, which we can leverage to evaluate the magnitude of our environmental perturbations. Differences between batch replicates reflect environmental factors that cannot be fully controlled experimentally, such as minor fluctuations in temperature and humidity, small inconsistencies in how the growth media were prepared, and biological stochasticity [51]. For each mutant, we calculated an absolute z-score for its fitness in a perturbed environment compared to its distribution of fitness across batch replicates (see Methods). We took the average z-score across all mutants for that environment as a measure of the magnitude of the environmental perturbations, as "perceived" by the yeast cells and reflected in their fitness effects. Fig 2D demonstrates that most perturbations within a base environment

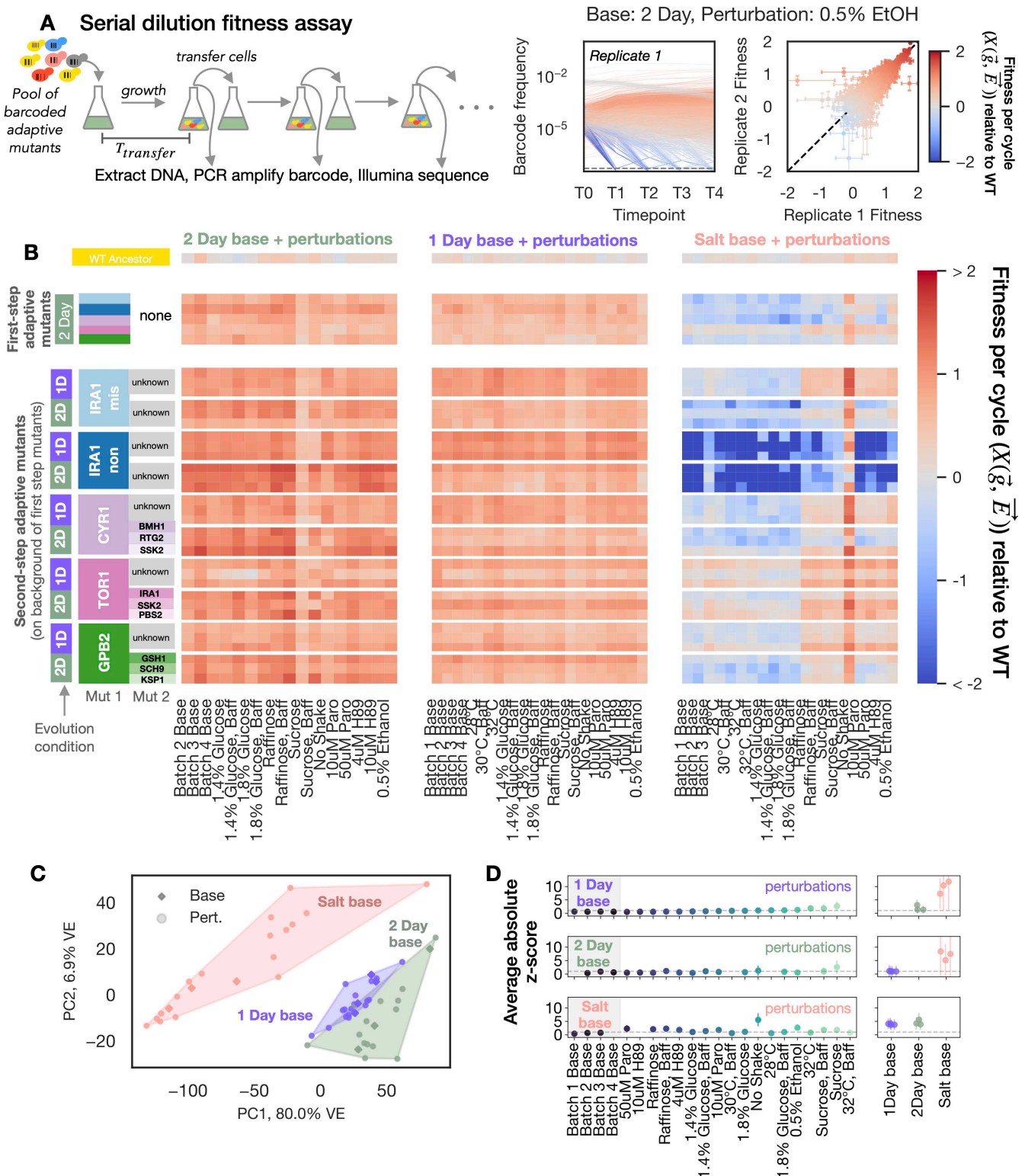

**Fig 2. Fitness assay of adaptive yeast mutants in three base environments with subtle perturbations.** (A) Fitness assay schematic: barcoded yeast strains with adaptive mutations from previous evolution experiments are pooled with WT ancestor (95% of pool). The mixture undergoes serial dilution every 24 or 48h ($T_{transfer}$). At each transfer, cells are preserved for DNA extraction, barcode amplification, and Illumina sequencing. This generates

precise strain-specific frequency trajectories over time, enabling high-resolution, replicable fitness measurements relative to the ancestor. **(B)** Heatmap displaying genotype-by-environment fitness matrix. Colors indicate fitness values, $X(\vec{g}, \vec{E})$. Columns represent environments (grouped by base), and rows represent three mutants from each ancestor-evolution environment combination. Top row shows ancestral wild type, subsequent rows show strains with 1–2 additional mutations from either 1 day ("Evo1D") or 2 day ("Evo2D") evolution environments. **(C)** PCA of environmental covariance matrix. Environments projected onto top two PCs, with unperturbed base environments as solid diamonds. Convex hulls outline all environments within each base. **(D)** Average absolute z-scores per environment, calculated relative to distribution of mutants' fitness across unperturbed base replicates. Perturbation colors follow 1 day base environment's ascending z-score order. Dashed line indicates z-score = 1. Right panels show other base environments' z-scores relative to focal base. The data underlying this figure can be found in https://zenodo.org/records/18175930.

typically result in average absolute z-scores around 1 when compared to their batch replicates, indicating their very subtle effects on fitness.

Altogether, this experimental design allows us to compare fitness variation of distinct groups of mutants due to identical environmental perturbations, applied to different base environments. Conveniently, we have groups of mutants that evolved in different base environments, lending us more power to dissect the role of short-term evolution in shaping the fitnotype space we detect, and to differentiate between pleiotropic expansion and pleiotropic shift when the mutants are moved to a distant base environment (Fig 1B).

## 2.2 Environmental perturbations generate base environment-dependent fitness effects, revealing ExE interactions

While Fig 2D suggests that the magnitudes of the same perturbation on different bases were similar, the "directions" might be different. For example, does adding 0.5% extra glucose to the media have a similar effect in a 24 hour dilution cycle and a 48 hour dilution cycle? And does the presence of salt disrupt the effect of additional glucose, if osmotic pressure exerts a more urgent challenge for the cells to respond to? Environment-by-environment interactions might indicate that fitness in different base environments is dominated by distinct challenges, lending support for the limiting functions model of fitness.

To isolate a perturbation's fitness effect on a particular mutant, we compute

$$\delta X_p^B(\vec{g}) = X(\vec{g}, (\vec{E}_B + \delta\vec{E}_p)) - \mu_B(\vec{g}) \tag{2}$$

where $X(\vec{g}, (\vec{E}_B + \delta\vec{E}_p))$ is the fitness of genotype $\vec{g}$ in an environment which is composed of perturbation $\delta\vec{E}_p$ on base environment $\vec{E}_B$, and $\mu_B(\vec{g})$ is genotype $\vec{g}$'s mean fitness across batch replicates of the base environment, $\vec{E}_B$. So $\delta X_p^B(\vec{g})$ is genotype $\vec{g}$'s fitness in the perturbed environment minus its fitness in the corresponding base, relative to the WT ancestor. Fig 3A shows $\delta X_{EtOH}^B$, the change in relative fitness due to adding 0.5% ethanol to each of the different base environments, for one strain with a mutation in the IRA1 gene, evolved in the 2 day home base from the WT ancestor. This mutant's fitness relative to the WT is shown in the top right inset panel for batch replicates of each base environment, representing the baseline from which $\delta X_p^B$ is calculated. Most perturbations cause smaller fitness effects than the baseline mutation fitness effect, consistent with Fig 2C, with several notable exceptions. The "No Shake" perturbation has little effect in the 1 and 2 days bases, but has a highly beneficial effect on the relative fitness of this mutant in the Salt base, suggesting an interaction between salinity and oxygenation. The addition of 0.5% ethanol is also beneficial on a Salt background, but deleterious on a 1 day background, and nearly neutral on a 2 day background.

Technical noise in the fitness measurement can also be a source of uncorrelated $\delta X_p$ across bases, so we compared base-to-base correlations with replicate-replicate correlations to distinguish noise from true ExE effects. For each biological replicate in one environment, we calculated $\delta X_R(\vec{g}) = \delta X_{p,R}^B(\vec{g}) - \mu_B(\vec{g})$, where we subtract each mutant's average fitness in the corresponding base from its fitness in the replicate; deviations from the 1:1 line for these comparisons can only be driven by noise, and provide a null for establishing a signal of ExE interactions. In other words, biological replicates act

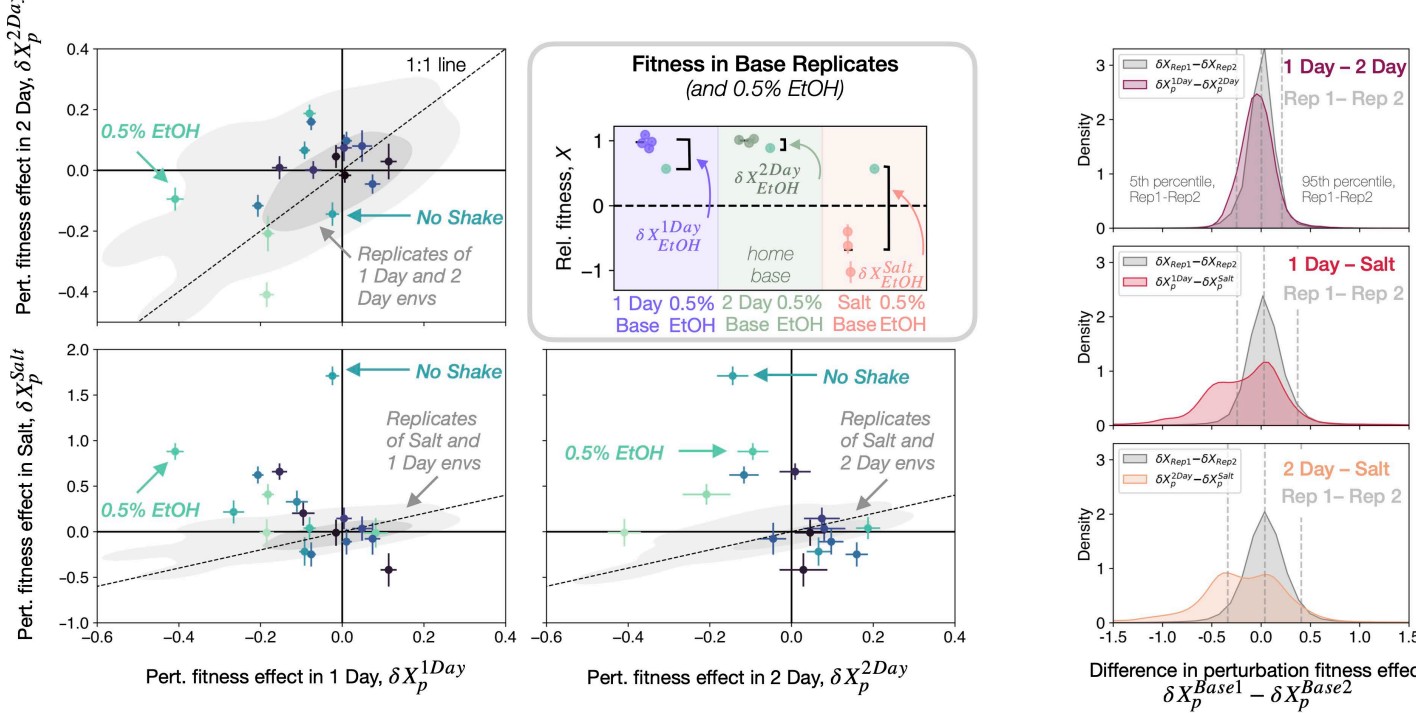

**A** Pairwise Comparisons of Perturbation Fitness Effects on Different Bases
IRA1 Mutant, evolved in 2 Day Base

**B** Differences in perturbation effects for all mutants

**Fig 3. Widespread environment-by-environment interactions hint at phenotypic novelty.** (A) Effect of environmental perturbation on one IRA1 mutant's fitness across different bases. Each perturbation is a blue-green circle, and error bars represent inferred standard error on the fitness measurement. Contour lines are the kernel density estimate for the replicate-replicate distribution, where we compare $\delta X^B_{p,R_1}$ and $\delta X^B_{p,R_2}$ for two biological replicates of the same environment. Perturbations that deviate from the 1:1 line more significantly than replicate-replicate pairs indicate the presence of ExE interactions. Inset: This IRA1 mutant's (evolved in 2 day base) raw fitness relative to WT in each base environment, and in 0.5% EtOH perturbation. $\delta X^{Base}_{EtOH}$ is the difference between relative fitness in perturbation and base. (B) Difference in perturbation effects across different pairwise combinations of bases, aggregated across all mutants. In gray, the distribution represents noise-driven differences, obtained from replicate-replicate $\delta X$ deviations. This is wider than the spread in panel A because it includes all mutants for both base environments represented in each panel. Vertical dashed lines represent 5th and 95th percentiles of the gray distribution. Colored distributions show pairwise comparisons of bases. 1–2 day comparison (top panel) shows a small, but significant difference between the two distributions. All distributions have been smoothed with kernel density estimation. Both comparisons with Salt (middle and bottom panels) have substantial density outside the gray distribution, suggesting the presence of many ExE interactions between Salt and the perturbations for many mutants. The data underlying this figure can be found in https://zenodo.org/records/18175930.

like the same perturbation on the *same* base, and thus should not be driving ExE interactions, whereas the same perturbation on *different* base environments might. The gray contour lines on the scatter plots in Fig 3A represent the distribution of $\delta X^B_{p,R_1}$ versus $\delta X^B_{p,R_2}$, the difference in $\delta X$ between biological replicates of each perturbations $p$ in the relevant base environments for the comparison at hand. We computed the Mahalanobis distance between each perturbation and the distribution of replicate-replicate correlations for each base comparison, and we found that for this IRA1 mutant, 12 out of 16 perturbations fall significantly outside any noise-driven ExE effects in replicate-replicate comparisons, implying that true ExE interactions are common.

We next aggregated the difference in $\delta X_p$ between two distinct base environments for all mutants and all perturbations. In Fig 3B, we plot the distribution of these differences, $\delta X^{B1}_p - \delta X^{B2}_p$. In gray, we again treat biological replicates as "perturbations" and plot $\delta X^B_{p,R_1} - \delta X^B_{p,R_2}$ for all pairs of biological replicates from the bases under consideration, which is narrow and centered around 0, compared to deviations across different base comparisons. The 1–2 day comparison

shows a small, but significant ($KS = 0.17$, $p < 10^{-4}$) difference in distribution from the replicate-replicate comparison, with 17% of comparisons demonstrating ExE interactions (FDR = 10%). The 1 day-Salt ($KS = 0.37$, $p < 10^{-4}$) and 2 day-Salt ($KS = 0.37$, $p < 10^{-4}$) comparisons are bimodal, with one peak corresponding to the technical noise-induced deviations, and another corresponding to significant ExE interactions across the two background bases. Here, 47% and 44% of comparisons, respectively, deviate significantly from the 1:1 line (FDR = 10%). Hence, we see strong evidence for individual mutants behaving unpredictably under ostensibly the "same" environmental perturbations. In Section S2, we explore the correlation in $\delta X$ for all mutants between pairs of environments, and find that certain perturbations show strong signs of ExE interactions. Overall, this suggests that genotype-by-environment-by-environment interactions may be crucial in shaping the fitnotype landscape we detect in each base environment.

## 2.3 Emergence of GxExE interactions in low-dimensional fitness landscapes

These environment-by-environment interactions naturally emerge in a simple model of a fitness landscape, motivated by empirical observations of low-dimensionality across environments [28–30]. To see this, we assume that fitness is mediated through a low-dimensional space of coarse-grained environmental variables, $\psi_k$, where $k = 1, 2, ..., D$. Therefore, we write fitness as a function of both genotype, $\vec{g}$, and these coarse-grained environmental variables, $\vec{\psi}$:

$$X(\vec{g}, \vec{E}) \rightarrow X(\vec{g}, \vec{\psi}(\vec{E})).$$

(3)

We model the fitness effect of an environmental perturbation by decomposing the environment into two parts: the base environment, $\vec{E}_B$, and the change in the environment due to the perturbation, $\delta \vec{E}_p$. If the perturbation is small, we expand the fitness function to write the change in fitness due to an environmental perturbation as

$$\delta X_p(\vec{g}|\vec{E}_B) = X(\vec{g}, \vec{\psi}(\vec{E}_B + \delta \vec{E}_p)) - X(\vec{g}, \vec{\psi}(\vec{E}_B))$$

$$\approx \sum_{k=1}^{D} \left.\frac{\partial X}{\partial \psi_k}\right|_{\vec{\psi}=\vec{\psi}(\vec{E}_B)} \cdot (\psi_k(\vec{E}_B + \delta \vec{E}_p) - \psi_k(\vec{E}_B)).$$

(4)

This equation, like Eq. (1), is a generalized version of the standard linear fitness models proposed in quantitative genetics literature, as well as an extension of the model used by Kinsler and colleagues to assign fitness to adaptive mutants via a low-dimensional latent phenotype space [28,52]. We can cast this equation for $\delta X_p$ into more standard notation, where the partial derivative plays the role of a "trait" or a "phenotype," and the $\vec{\psi}(\vec{E}_B + \delta \vec{E}_p) - \vec{\psi}(\vec{E}_B)$ term is a selection gradient. To simplify our notation, we write

$$\delta X_p(\vec{g}|\vec{E}_B) \approx \sum_{k=1}^{D} \phi_k(\vec{g}|\vec{E}_B)\beta_k(\vec{E}_p|\vec{E}_B),$$

(5)

where $\phi_k(\vec{g}|\vec{E}_B)$ is an effective phenotype that depends on the genotype and base environment, and $\beta_k(\vec{E}_p|\vec{E}_B)$ is the selection gradient that depends on the base environment and perturbation.

Equation (4) shows that we should generically expect gxExE effects via the $\delta X_p$ metric, unless the base environments are so close to each other that the $\phi_k(\vec{g}|\vec{E}_B)$ is actually base-independent, and the weights $\beta_k(\vec{E}_p|\vec{E}_B)$ are the same across bases. Thus, the observations in Fig 3 of ExE interactions are consistent with this simple model, since we chose our base environments to be far from one another. We can infer that they are distant enough to induce widespread gxExE interactions. Even so, if the environmental *perturbations* are small, the linear approximation in Eq. (5) is still a good one. We thus cull the Salt + No Shake perturbation (which was not subtle according to Fig 2D), and henceforth use a linear fitness model

as in Eq. (5) to probe the quantitative properties of environmental dependence of phenotypic landscapes. In particular, we can use Eq. (5) to infer $D$ for each base environment, allowing us to differentiate between the pleiotropic expansion and pleiotropic shift models. Additionally, we can use this model to determine whether ExE interactions arise from a unique set of $\vec{\phi}$ vectors in each base, or if instead there is a shift in the weights, $\vec{\beta}$, shedding light on how the limiting functions that determine fitness might change across different base environments.

## 2.4 Environmental shifts away from home base do not induce pleiotropic expansion

Having established that environment-by-environment interactions are prevalent in our data, we next sought to distinguish whether adaptive mutants fit the pleiotropic expansion or pleiotropic shift models presented in Fig 1, using a linear model of fitness as in Eq. (5). Kinsler and colleagues proposed that adaptive mutations could have large fitness effects *and* be pleiotropic because they are modular near their evolution condition [28]. In other words, a set of adaptive mutations might look the same in environments that are very similar to their evolution condition because they affect fitnotypes in the same way. But there are no guarantees for low-dimensional behavior amongst this set of adaptive mutants when their fitness is measured in new environments that are dissimilar from the evolution condition. Under this model, we would expect the fitness of a set of adaptive mutants to be well-approximated by a low-dimensional model in the home base (and subtle perturbations). But as the environment shifts, previously hidden phenotypic effects may reveal themselves, and tradeoffs may emerge. Does this manifestation of hidden pleiotropy tend to drive higher-dimensional behavior of adaptive mutants in foreign environments?

To test this, we first focused on the adaptive mutants that arose on the WT background in the 2 day base environment. We partitioned our environments by base and asked whether the same set of mutants yielded the same low-dimensional behavior around each base environment. Specifically, we wondered whether the 2 day base would look different from the other two bases because it has the special distinction of being the evolution environment for this set of mutants. Under the "pleiotropic expansion" model, we expect the dimensionality of the mutant set to appear higher around Salt and 1 day than 2 day.

To infer the dimensionality of the latent space, we did SVD on the $\delta\mathbf{X}$ matrix of perturbation effects on fitness in each base separately, and quantified the variance explained by each dimension. By definition, the first $k$ SVD components generate the $k$-dimensional linear model with the lowest reconstruction error, and as $k$ approaches the rank of the input matrix, the variance explained approaches 100% [53].

We compared the fraction of variance explained by each component across the three different base environments. Due to different levels of noise and measurement resolution, we have different limits of detection for each base, though the 1 day and 2 day limits of detection nearly coincide. We compared the number of components that fall above the variance explained by the limit of detection. We obtained this limit by running SVD on 1,000 folds of a noise-only matrix, filled with 0-centered Gaussian random variables, with standard deviation equal to inferred standard error due to measurement noise in the fitness assay. As shown in Fig 4A, the $\delta\mathbf{X}$ of the mutants that evolved in the 2 day base are well-approximated with a 4-dimensional model in environments belonging to the 2 day base. When the same set of mutants is measured in the same set of environmental perturbations on the 1 day base, 3 components fall above the limit of detection. And the most distant environment, Salt, is also well-captured by a 3-dimensional model, though the scree plot in Fig 4A is steeper and three dimensions explain more variation than in the other two bases. Even so, for this particular set of mutants, we find that dimensionality in the evolution base (2 day) does not substantially differ from the non-evolution condition bases, and in fact is slightly higher by this particular metric. We explore several other methods to infer dimensionality in the supplemental information, and these results persist (Section 4 in S1 File, S4 Fig).

But this set of mutants effectively yields only two comparisons–home base with two other bases. To make a claim about the typical behavior of adaptive mutants in divergent environments, we need to make more comparisons of dimensionality between home and away base environments. Here, we leverage the second-step mutants from subsequent evolution

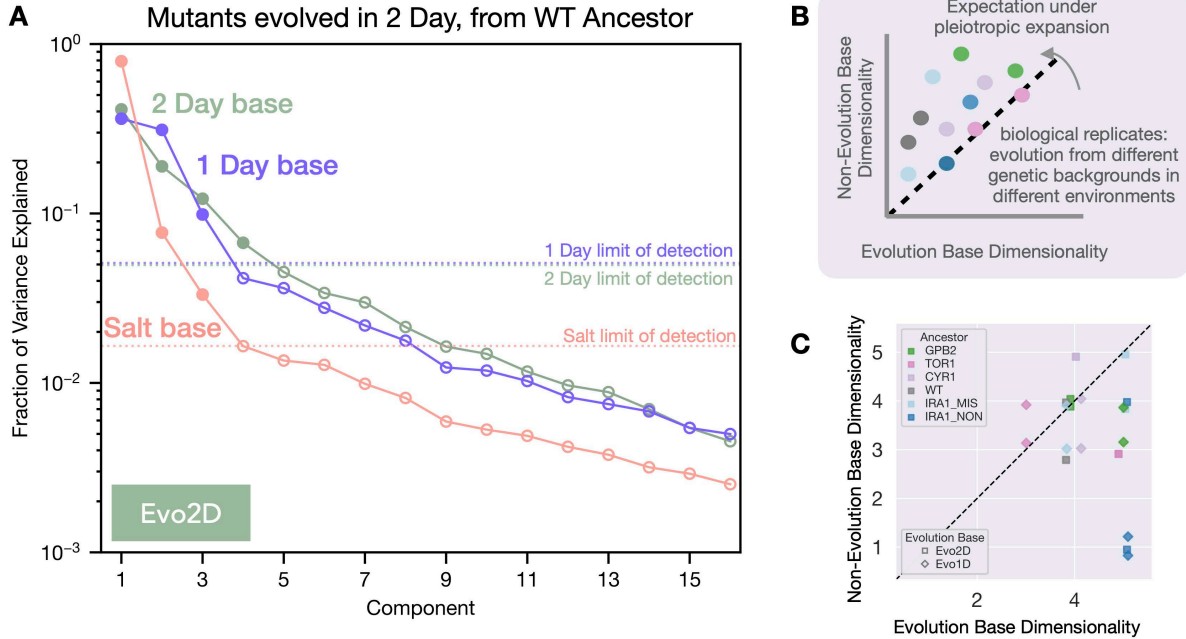

**Fig 4. Pleiotropic expansion model is not supported by dimensionality reduction across all bases. (A)** Fraction of variance in $\delta X$ explained by component for each of the three environmental bases. Points are colored if they fall above limit of detection. Limit of detection was obtained by doing SVD on 1000-folds of noise-only matrices, using inferred measurement error on fitness as the standard deviation for the randomly drawn fitness in each fold. These mutants evolved in the 2 day base, and have 4 dimensions above the limit of detection in 2 day, 4 in Salt, and 3 in 1 day. **(B)** If pleiotropic expansion were true, we should expect systematically higher dimensionality in a non-evolution condition base than an evolution condition, amongst a set of biological replicates (i.e., independent evolution experiments). **(C)** Dimensionality in evolution environment and non-evolution environments for different "biological replicates" - sets of evolved mutants from different ancestral backgrounds, and in different environments (jitter added). We find no evidence for pleiotropic expansion. The data underlying this figure can be found in https://zenodo.org/records/18175930.

experiments, represented by the bottom rows in Fig 2B. These mutations arose in both the 2 day base and 1 day base, so we now have two evolution conditions to compare, and to lend more evidence for or against the pleiotropic expansion hypothesis. These sets of mutants serve as biological replicates–they represent multiple instances of diversification from different genetic backgrounds, in two different environments. For each set of mutants, we quantify dimensionality in their evolution environment, and the other two bases. We visualize these comparisons as a scatter plot, with dimensionality in evolution condition on the x-axis, and dimensionality in distant environmental base on the y-axis. Fig 4B shows a schematic for our expectation under the pleiotropic expansion model, whereby dimensionality in the nonevolution base is systematically higher than the evolution base, for the same set of mutants. Fig 4C shows the empirical data, in which we inferred dimensionality for each genetic background and evolution condition in this study. Each point is a set of adaptive mutants, whose dimensionality is quantified using the approach from Fig 4A. We include the scree plots of the fraction of variance explained by each dimension for all other mutants in the supplemental information (S3 Fig).

Overall, we find no evidence for the pleiotropic expansion hypothesis. In general, there is no clear relationship between dimensionality and whether a group of mutants evolved to the base or not. In fact, the scatter plots indicate that the evolution condition appears to be higher-dimensional more often than the other bases, but more replicates would be necessary to confidently make such a statement. We can claim, however, that a similar number of dimensions is necessary to build fitnotype maps for each set of mutants, regardless of where they evolved. Additionally, we quantify the dependence of inferred dimensionality on number of environments included in the input matrix in the supplemental information. None of these additional analyses change our conclusions. This allows us to reject the pleiotropic expansion hypothesis, and turn

our attention to quantifying how fitnotype space shifts across distinct base environments. In the rest of this work, we aim to characterize how different base-dependent limiting functions influence the way fitnotypes in different base environments map onto each other.

## 2.5 Characterizing pleiotropic shift across base environments via fitness prediction

We have shown that each base environment reveals a similar number of fitnotypes for all sets of mutants, suggesting that low-dimensionality is not specific to the evolution condition. We next wondered the extent to which different fitnotype spaces overlap. Are these base environments limited by the same functions?

It is difficult to answer this question directly because fitnotypes are latent variables, so instead we use predictive power as a proxy for fitnotype overlap (see Section S1 for more details). Prediction error is invariant to the choice of basis in matrix factorization, so it is a useful metric for this task. Fig 5A presents our approach. First, we select one base as the "training base," and hold out a single perturbation $i$. We do SVD on the training base (excluding $i$) to discover fitnotypes in this base for all mutants. Then, we use the left singular vectors, here playing the role of the fitnotypes $\vec{\phi}$, as features in a linear regression model, and fit new "weights" on the fitnotypes to predict $\delta X$ in the held-out perturbation $i$, obtaining a leave-one-out cross-validation error. Similarly, we can fit weights on the training base fitnotypes to predict $\delta X$ from any perturbation, including those from other bases, yielding prediction error for "test base" perturbations. We aggregate prediction error for all held-out perturbations within the training base, and all perturbations from the other two test bases, to quantify the cross-validation predictive power of each fitnotype to compare. Importantly, the cross-validation error even within the "training" base is still a test error, because that perturbation was left out of the initial fitnotype discovery. We are essentially asking: if we allow our model to freely re-weight training fitnotypes to best predict $\delta X$ in a new base, how close can we get to within-base predictability?

We have different expectations for this predictive power across bases under different models of how fitnotype spaces map onto one another, as shown in Fig 5B. First, we note that even within the training base there will be unexplained variance due to noise and/or nonlinearities that cannot be captured by a 6-dimensional linear model. We represent this unexplained variance in gray. If fitnotypes are fully orthogonal (Fig 5B, i.), we do not expect that any amount of re-weighting will allow us to fit $\delta X$ in the testing base, so the total variance explained by all training fitnotypes (green bars) will be negligible in the test base. We indicate the maximum achievable prediction had we inferred fitnotypes directly from the test base with a "prediction ceiling," and highlight the gap in predictive power using the test base color and white hashing. This scenario corresponds to a strong limiting functions model, where fitness-relevant functions are entirely distinct across environments and $\phi$ vectors are orthogonal, clearly yielding ExE interactions (Results C, Eq. (5)). At the other extreme (Fig 5B, iii), fitnotypes could fully overlap between bases. Although weights ($\vec{\beta}$) may differ, the underlying functions remain the same, leading to comparable predictability across bases because linear regression allows the flexibility of re-weighting. ExE interactions can emerge from this model, but they would be driven by changing weights rather than fitnotypes. Between these extremes (Fig 5b, ii), environments may harbor some of the same fitnotypes, but not all. These shared limiting functions yield moderate predictive power, but the distinct functions also create a substantial gap in predictability between the training and test bases. We confirm these theoretical expectations with simulated fitness data in the supplemental information (Section 6 in S1 File, S6 Fig).

In Fig 5C, we compute the variance explained by different fitnotypes as presented above, for all mutants. We find partially overlapping fitnotypes between all base environments. As anticipated, the limiting functions between 1 and 2 day appear to be more similar, as the gap in prediction power between 2 and 1 day is smaller than between 2 day and Salt (Fig 5C, left panel). To aid in the interpretation of the size of the predictive gap, we performed the same regression procedure described above on a matrix constructed only from the difference in $\delta X$ between biological replicates, $\delta X_R$. We find that the maximum variance explained for held-out perturbations within a training base is for the 2 day base, at around 11.2% variance explained, and for a testing base, 2 day fitnotypes explain 6.4% of the variance in 1 day perturbations.

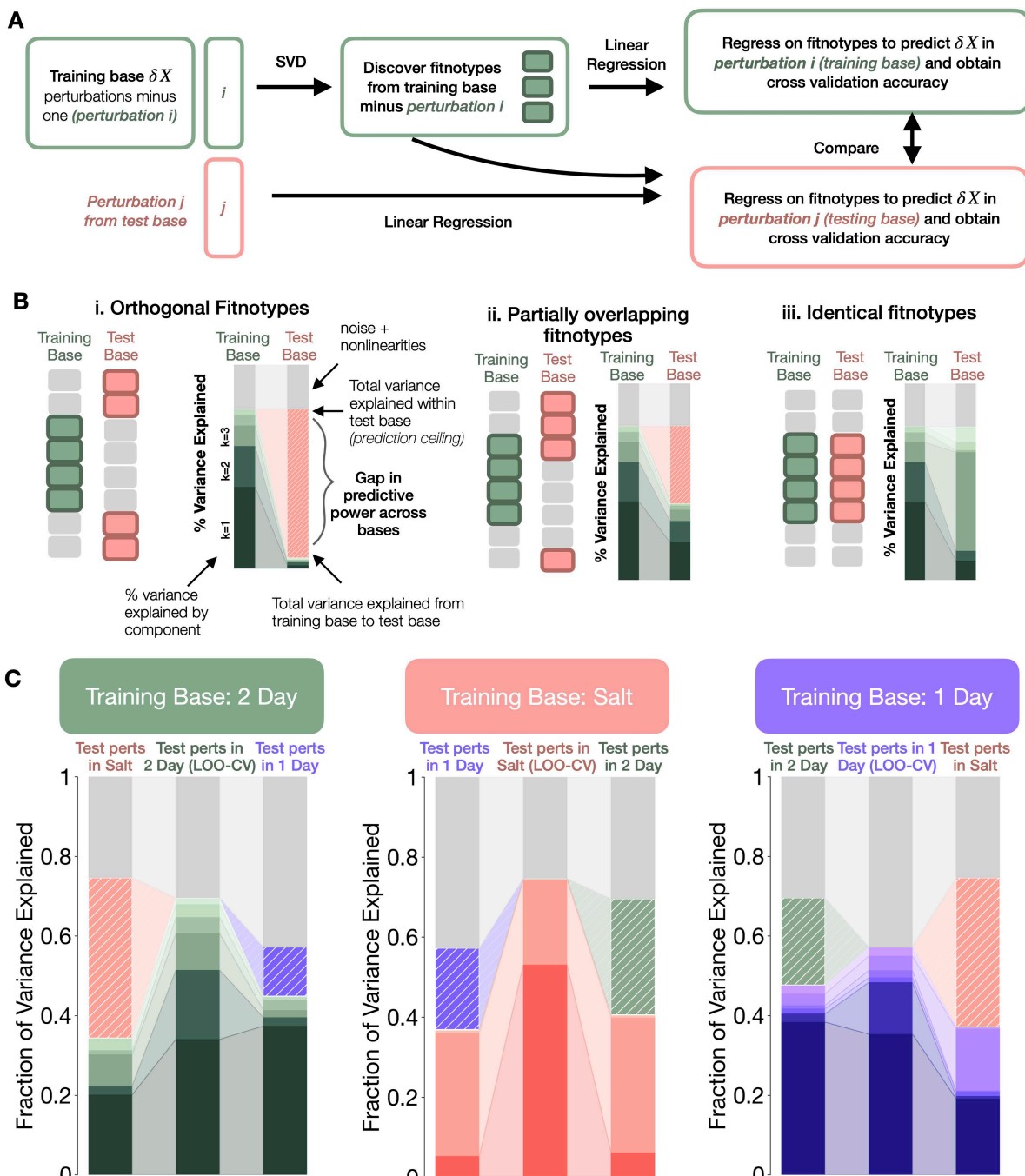

**Fig 5. Predicting perturbation fitness effects within and across environments reveals structure of limiting functions. (A)** Prediction workflow: We partition training data to extract base-specific fitnotypes via SVD, use these as features in a linear regression to predict fitness changes in held-out perturbations, and quantify variance explained within and across base environments. **(B)** Expected prediction patterns in different scenarios: (i) orthogonal fitnotypes between bases prevent cross-base prediction; (ii) partially overlapping fitnotypes allow some cross-base prediction with characteristic gaps (pink dashed section is missing test base variance, gray is unpredictable variance); and (iii) identical fitnotypes yield equivalent predictive power across bases. **(C)** Our data reveal partially overlapping fitnotypes, supporting scenario (ii), for these mutants and environments. The data underlying this figure can be found in https://zenodo.org/records/18175930.

The maximum gap from noise alone is around 10% of the variance, as Salt fitnotypes only explain 0.8% of variance in 2 day. This serves as a reasonable estimate for how large the "gap" in predictability can be purely due to noise and finite size effects. We include analogous bar plots for these noise-only results in the supplemental information, in S11 Fig. All of the gaps in predictability in Fig 5C exceed this threshold, suggesting that there is at least some unique fitnotype space in each base environment.

We also observe some reweighting of fitnotypes in different bases. For example, the fifth 1 day fitnotype contributes little to predict $\delta X$ in the training base, but grows to explain nearly 20% of the variance in Salt (Fig 5C, right panel). Note too that we fit a 6 dimensional model in the linear regression (Fig 5A) to lend extra flexibility, but we see evidence for much lower dimensionality, particularly in Salt, which is dominated by just two fitnotypes. In the supplemental information, we show similar prediction analyses for individual sets of mutants, and predictive power for individual test perturbations within the bases (S7 and S8 Figs, Sections 7–8 in S1 File). We also show in the supplemental information that a genotype-fitness model that includes fitnotypes and flexible re-weighting outperforms other low-dimensional models (i.e., simple gene-level predictions) for predicting $\delta X$ (S9 Fig, Section 9 in S1 File).

Taken together, these findings combine to strongly support the pleiotropic shift hypothesis, with partially overlapping fitnotypes across different bases. We see direct evidence for phenotypic variation that remains hidden in certain bases, but that is revealed in others. Additionally, within the overlapping phenotypic space, the traits that most strongly contribute to local fitness often shift. It is worth noting that the inputs to our model for each base environment are, on the surface, identical. The same genotypes and the same physical and chemical changes to the base environment make up the rows and columns of the input matrix. Therefore, the heterogeneity in fitnotype space across base environments is entirely attributable to how the base environment constrains the fitness function. This lends support for the idea that fitness in a given environment is dominated by a small number of limiting functions, and highlights the importance of environmental context in the inference of genotype-phentoype-fitness maps.

## 3 Discussion

In this study, we aimed to understand how genotype-phenotype-fitness maps shift across environments. Building such maps has been a difficult task for several reasons. (i) Genotype space is vast, (ii) it is unclear which phenotypes are important to fitness, and (iii) environments can modulate both the genotype-to-phenotype mapping, and phenotype-to-fitness mapping. Previous work tackled the initial question of mapping phenotype space for a group of adaptive mutants near their evolution environment, using a top-down linear model of fitness variation across subtly perturbed environments [28]. This technique yields a map of abstract "phenotypes" that are revealed through their effect on fitness that Kinsler and colleagues (2020) termed "fitnotypes." In generating full genotype-fitnotype-fitness maps around distinct base environments, which are themselves distant from each other, we were able to explore the nature of local and global pleiotropy for adaptive mutants and distinguish between two models of how pleiotropy might manifest.

One hypothesis, the pleiotropic expansion hypothesis (Fig 1A), is that the dimensionality of a model capturing the fitness variation of adaptive mutants is small near their evolution condition, but grows dramatically around distant bases. We did not see evidence supporting this model: we reliably found instead that the dimensionality of our models was consistently low across bases, regardless of whether they evolved in the base environment or not. This allowed us to reject the "pleiotropic expansion" model.

We instead found strong evidence for the alternative, the pleiotropic shift model (Fig 1B). In this model, genetic mutations do generate a pleiotropic collection of phenotypic effects, but the dimensionality of the phenotypic variation that affects fitness remains limited even in environments that are distant from the evolution condition. This pattern is consistent with a "limiting functions" model of fitness. In any particular environment, there is a small number of key limiting functions that can be altered to affect fitness. These are the functions that determine the cells' ability to surmount limiting challenges and thrive. Fitness variation is thus dominated by this small set of functions, suggesting that phenotypic diversity

in an adapting population is constantly being projected down onto a few key dimensions. The choice of these dimensions changes as the environment shifts. This model bears resemblance to Fisher's Geometric Model, in which organisms adapt in a multidimensional trait space, typically with a quadratic or Gaussian fitness function whose peak represents an environmental optimum [1,10]. An exploration of connections between a "limiting functions" model and a sparse, anisotropic Fisher's geometric model would be a rich direction for future theoretical work.

How might such a pleiotropic architecture of limiting functions arise? Similar questions have been explored extensively in the context of nutrient availability [41,44,45]. If cellular growth is a function of available nutrients, many different nutrients may be required. However, at any given time, growth will be constrained by just one of these—the limiting nutrient—while the others remain in excess. This principle is well understood in both ecology and chemistry; for example, in a chemical reaction, the limiting reagent determines the reaction's progress. Fitness is a more complex function than a simple chemical reaction, but at its core, it is governed by a series of biochemical processes that drive growth. Differences in these processes create differences among cellular performances, manifesting as fitness advantages or defects. It is therefore reasonable to extend the concept of limiting nutrients or reactants to the level of fitness. Importantly, the key constraint on fitness may not always be nutrient deficiency. It could be the need to withstand extreme heat, buffer against extreme pH, or overcome some other environmental challenge. Ultimately, a cell "perceives" its environment as a set of challenges, but in any given moment, closely related cells will be limited by only one or a few of these challenges. After all, optimizing growth rate is irrelevant if the cell is dying from osmotic stress.

We are presented with a range of possibilities for how environments might vary under this model. Two distant environments might present entirely orthogonal selection pressures to a population. In a simple case, one environment might have an abundance of a fermentable carbon source for the yeast, so optimizing growth rate or lowering lag phase for fermentation metabolism the best function to optimize in a fitness competition. But a different environment might only have nonfermentable carbon sources, and so changes to fermentation metabolism will have little effect on fitness in this second environment. Any pleiotropic effects from the first environment that bear on respiration metabolism will be revealed, and differentiate between the various mutants. Imagine another scenario, in which both of these media are placed in an incubator at 37˚C. Now, while the base media are still "orthogonal" metabolically, the yeast is most concerned with surviving the heat shock, and so the limiting function is heat survival in both environments. Any variation in that phenotype will dominate the relative fitness between mutants in both environments, so we will infer these underlying fitnotypes to be fully overlapping. Conceptualizing environments in this manner presents an intriguing possibility for understanding environmental similarity, which has historically been difficult to define. The notion of overlap in limiting functions could be a useful organism-specific metric for parameterizing environmental distance, from the perspective of competitive fitness.

One natural question to emerge is why there seems to be overlap in the phenotypes affected in our pool of mutants, and the limiting functions that are important in our distant, nonevolution base environments. *A priori*, it is not clear that variation in the hidden phenotypic effects from a group of mutants will necessarily be relevant to the limiting functions in an arbitrary environment.

The most obvious possibility is that perhaps the bases are not in fact sufficiently distant. All three are lab environments with glucose-limited media, so while we do observe many differences in fitness, perhaps we are constraining the environment space more than we hoped. Or perhaps the very fact that yeast cells can survive in all these environments necessarily acts as a built-in constraint on fitness assays and growth, meaning that the range of possible environments will always be limited. So it is interesting to speculate: what will we learn when we apply many more perturbations to a very large number of base environments? Perhaps we will discover that there are endless ways to project phenotypic diversity onto fitness, and that at a certain point the collection of genotypes will become the limiting variable because phenotypes are indeed low-dimensional [54–56]. Or perhaps we will discover that environment space, constrained by survivability, is actually small, and consistently mappable. This extension also addresses a limitation of our approach, which is the diversity of perturbations. In order to fully explore fitnotype space, one would want to choose maximally diverse subtle perturbations, so

as to vary fitness in all possible dimensions. In this work, we chose sets of perturbations which themselves are correlated (e.g., different glucose concentrations), which could plausibly lead to the observed low-dimensionality, though perhaps not the lack of full overlap in fitnotype space. These extensions of both the diversity perturbations and the number of base environments to more fully tile environment space provide a promising avenue for future work.

Still another possibility for why we tend to detect limiting functions across the three base environments is that single bouts of evolution can exhaustively explore all phenotypic diversity, due to the tight integration of cellular and genetic networks, such that any possible important phenotype is already present in an adapting population of sufficient size [2,48,57,58]. This might suggest that there are indeed few phenotypes that can be modified to affect fitness in this set of environments. Perhaps a core set of physiological phenotypes, such as lag time, growth rate, diauxic shift efficiency, etc, are determining fitness in all lab environments. Our observation of novel fitnotypes in different bases may be a shift in their relative weights such that some become undetectable in certain bases, while the overall number remains small. This consideration motivates future work in the direction of identifying these phenotypes, perhaps through a set of targeted physiological measurements. The low-dimensionality hints that such measurements may not be overly cumbersome, and that these core physiological phenotypes might be learnable. Or, perhaps we are biased by our genotype set. Most of the mutations in our collection are either autodiploidization events, which affect the entire genome, or in signaling pathways, which are known to be especially pleiotropic [5,49]. These mutants are also atypical in that they harbor large beneficial fitness effects, despite their genetic proximity. What might we find by applying similar environmental perturbations and measuring the relative fitness of a different set of genotypes? Perhaps a library of "random" mutations from transposon insertion, for example, would generate more novel behaviors across environments. It would also be enlightening to compare our results to a set of genotypes that have adapted for much longer than ours, either from experimental evolution with the LTEE *E. coli* lines for example, or as a result of evolution in natural settings by looking at a large collection of natural isolates that represent the extent of standing genetic diversity within a species [59–61].

The limiting functions model naturally predicts that the fitness effects of environmental perturbations can often be heavily dependent on the base environment. If base environments present distinct challenges, the projection of fitness onto these key dimensions will be different, even as a result of the same underlying physical and chemical changes. Thus, idiosyncrasy in $\delta X$ effects across bases is to be expected. However, this idiosyncrasy does not mean that prediction writ large is hopeless. The fact that dimensionality remains constrained locally in any base environment raises the possibility of practical applications, because the total space of fitness variation is smaller than we might initially expect.

More broadly, the limiting functions model offers insight into how evolution might proceed over successive environmental epochs. First, a clonal population accumulates multiple adaptive mutations that compete with each other for fixation. All of these adaptive mutations confer a fitness benefit relative to the ancestor, and tend to affect similar phenotypes that matter to fitness in their environment, but they harbor a reservoir of hidden phenotypic diversity. When the environment shifts and a new epoch begins, a different set of phenotypes will be important for fitness, reshuffling the relative fitness of each mutant. The pool of phenotypic diversity amongst a group of adaptive mutations is large when integrated across multiple environments, but is never so large in a single epoch that mutants behave in completely idiosyncratic ways. Environments present limiting challenges that impose low-dimensional structure on phenotype space, and dominate the fitness response. This low-dimensionality provides some hope that evolutionary prediction is possible, and brings us one step closer to the ambitious goal of building a universally relevant genotype-phenotype-fitness map.

## 4 Methods

### 4.1 Experimental procedure for barcoded fitness assay

**4.1.1 Experimental overview of fitness assay.** We performed fitness assays as previously described [28,49,62]. We pool all barcoded strains and compete them together against a reference strain (in this case the wild type ancestor, WT, represented at 95% of the pool at the beginning of the competition) by growing them for a predetermined transfer

duration ($T_{transfer}$) in 100 mL of glucose-limited media. We grew the cells in incubators set at temperatures determined by environmental prescription, for either 24 hours or 48 hours per cycle. All environments except "No Shake" were rotated at 223 rpm. At the end of each cycle, after the cells have grown in direct competition, we transfer 400 μL of saturated media into fresh media, or roughly $5 \times 10^7$ cells, preferentially sampling lineages that have grown to higher frequency over the course of the cycle due to increased fitness relative to the mean population fitness. We dilute the cells at a 1:250 ratio, corresponding to roughly 8 generations of cell division over the course of the growth cycle, and we perform 4 total dilution cycles. At each transfer, we save the rest of the untransferred cells in sorbitol for DNA extraction and PCR amplification of the barcode region. We sequence 5 timepoints to get estimates for the frequency of each barcoded lineage, and use the log-fold change in frequency to infer the relative fitness of each strain with respect to the WT ancestor, correcting for the changing population mean fitness. With these methods, in each environment we were able to infer the relative fitness of thousands of strains. We perform fitness assays in multiple environments (see S1 Fig for details of environments).

**4.1.2 Yeast strains.** All strains in this study are of genetic background Mat*a*, ura3Δ0::Gal-Cre-KanMX-1/2URA3-loxP-Barcode-1/2URA3-HygMX-lox66/71. Mutant strains come from the evolution experiments described in [46–48]. The mutations that differentiate the barcoded strains from the WT arose in several evolution experiments via serial dilution. One study, Levy and colleagues, generated an initial set of adaptive mutants from a single WT ancestor that evolved in glucose-limited media with 48 hour dilution cycles ($T_{transfer}$ = 48 hr) over 160 generations [46]. Each mutant typically harbors one putatively causal genetic variant that improves fitness relative to the ancestor in this environment. Most of these mutations were either auto-diploidization events, or single nucleotide changes in genes in the Ras/PKA and TOR/Sch9 pathways, both nutrient-sensing pathways [49]. In subsequent studies, four of the Ras/PKA mutants and one TOR mutant were re-barcoded and further evolved in two environments: the original 48 condition ("2 Day") and a 24 hour condition ("1 Day") [47,48]. The cells complete fermentation of the glucose at around 20 hours and switch to nonfermentable carbon sources through respiration afterwards, so the 1 Day condition greatly limits the time the cells spend in respiration, amounting to a significant environmental shift despite unchanged media [62].

**4.1.3 Growth environments.** All of our growth environments are based on M3 minimal, glucose-limited media. Our environments are defined by the media in the flask, additional components added to the media, the shape of the flask, the shaking status, and the temperature of the incubator. Due the scale of our experiment, we grouped our environmental perturbations into 4 batches, each with 3 base conditions and between 4 and 5 perturbations. This meant a total of 12–15 environments per batch. We had two biological replicates per environment. The first batch was temperature perturbations, the second batch was glucose gradients, the third batch was additional carbon sources, and the fourth batch was drugs and ethanol. See Supplemental Information for a full list of environments (S1 Fig).

**4.1.4 DNA extraction.** We extracted genomic DNA according to a similar procedure in [28]. Briefly, we thawed 400 μL of cells for each sample. We spun down the cells at 3,500 rpm for 3 min, and discarded the supernatant. We washed the cells by adding 400 μL of sterile water and spun them down at 3,500 rpm for 3 min, and discarded the supernatant. We resuspended the pellet in 400 μL of extraction buffer, and incubated at 37 °C for 30 min. Then, we added 40 μL of 0.5M EDTA, and vortexed. Then, we added 40 μL of 10% SDS, and vortexed. We next added 56 μL of proteinase K, and vortexed very briefly. We incubated at 37 °C for 30 min, and put the samples on ice for 5 min. We then added 200 μL of 5M potassium acetate, and manually shook the tubes. We incubated on ice for 30 min. Then, we spun the tubes for 10 min at maximum speed (13,000 rpm), and transferred supernatant to a new tube with 750 μL isopropnaol, and rested it on ice for 5 min. We spun it down for 10 min at max speed and discarded supernatant. We eashed twice with 750 μL of 70% ethanol, vortexing very briefly and spinning at maximum speed for 2 min, then discarding liquid. We resuspended in 50 μL Tris pH 7.5, and left it on the bench overnight if the pellet was not fully resuspended. We next added 1 μL of 20 mg/mL RNase A and incubated at 65 °C for 30 min. To digest the ancestor (so that we do not sequence our high-abundance

reference strain), we added 5.5 $\mu$L of Cutsmart buffer and 1$\mu$L of ApaL1 restriction enzyme to each tube. We incubated the tubes at 37 °C overnight, and quantified with Qubit.

**4.1.5 PCR amplification of barcode locus.** To amplify the barcode region for sequencing, we perform two steps of PCR. For the first step, our goal is to tag individual molecules with UMIs and include first step primers to allow for combinatorial indexing. We divided each sample into 8 separate PCR tubes in order to eliminate the influence of PCR jackpotting. For a single set of 8 reactions, we used 200 $\mu$L HotStartTaq Polymerase master mix, 8 $\mu$L each of forward and reverse primers (10 $\mu$M), 118 $\mu$L nuclease-free water, 16 $\mu$L of 50 mM MgCl2, and 50 $\mu$L of genomic DNA. We ran 3 cycles of the following program.

1. 94 °C 10 min

2. 94 °C 3 min

3. 55 °C 1 min

4. 68 °C 1 min

5. Repeat steps 2–4 for a total of 3 cycles

6. 68 °C 1 min

7. Hold at 4 °C

We did a clean-up (standard DNA purification procedure) of the step 1 PCR product, and then performed step 2 PCR. We split our template DNA into 3 reactions, and for a set of 3 reactions, we used 45 $\mu$L template, 65.5 $\mu$L nuclease-free water, 30 $\mu$L Q5 buffer, 2.5 $\mu$L 10uM Forward and Reverse Nextra primers, 3 $\mu$L 10mM dNTPs, and 1.5 $\mu$L Q5 polymerase. We then ran the following program for the second step of PCR, which amplifies for 20 cycles.

1. 98 °C 30 s

2. 98 °C 10 s

3. 62 °C 20 s

4. 72 °C 30 s

5. Repeat steps 2–4 19 times

6. 72 °C 3 min

7. Hold at 4 °C

We did a sample clean-up using standard DNA purification techniques, and quantified with Qubit.

**4.1.6 Sample pooling and amplicon sequencing, and data processing.** We pooled our samples and sent them to Admera Health for quality control, bead cleanup, and sequencing. We sequenced on HiSeq and Novaseq S4. We processed our data in exactly the same way as [28] to obtain barcode counts and frequencies for each sample, from raw sequencing reads.

## 4.2 Analysis

**4.2.1 Fitness inference.** We adapted the fitness inference method introduced in Ascensao and colleagues 2023 [50], and closely follow their derivation of maximum likelihood fitness estimation below. We want to infer the fitness of a barcoded yeast strain from the read count information. Read counts for a particular barcode are genereteed through a sequence of noisy processes, such as stochastic growth, bottle-necking, DNA extraction, PCR

amplification, and sequencing. Each of these individual processes a counting process, so we can model the read count of barcode *i* at time point *t* as a negative binomial random variable, which essentially allows us to think about the actual value of the read count as being sample from an over-dispersed Poisson distribution, where the variance is larger than the mean:

$$r_{i,t} \sim \text{NB}(\mu_{i,t}, c_t),$$

(6)

where the expected value of the read count is

$$\langle r_{i,t} \rangle = \mu_{i,t} = R_t f_0 e^{(s - \bar{x}_t)t}$$

(7)

and the variance is

$$\text{var}(r_{i,t}) = c_t \langle r_{i,t} \rangle.$$

(8)

There are several parameters we need to infer before we can try to infer *s*. In order to calculate the expected value of the read count, $\langle r_{i,t} \rangle$, we must have an estimate of the mean fitness of the barcoded population with respect to the ancestor, $\bar{x}_t$, which we can infer from the trajectories of known neutral barcoded lineages. For a set of neutral barcodes, we have (by definition):

$$f_t^{(n)} = f_0^{(n)} e^{-\bar{x}_t t},$$

(9)

$$\bar{x}_t = -\frac{1}{t} \log \frac{f_t^{(n)}}{f_0^{(n)}}.$$

(10)

We can actually calculate this mean fitness, with the caveat that read counts are somewhat noisy estimates of the underlying frequency. Also, since we assume all the barcodes that identify neutral lineages should have identical fitnesses (s = 0), we can take the sum of all neutral barcodes at a given time point, $R_t^{(n)}$, to calculate the mean fitness.

$$\bar{x}_t = -\frac{1}{t} \left[ \log \frac{R_t^{(n)}}{R_t} - \log \frac{R_0^{(n)}}{R_0} \right],$$

(11)

where

$$R_t^{(n)} = \sum_{i \in \text{neutral}} r_{i,t}.$$

(12)

Alternatively, we can calculate the mean fitness within time intervals, where we have

$$\bar{x}_{j,k} = \sum_j^k -\frac{1}{k-j} \left[ \log \frac{R_k^{(n)}}{R_k} - \log \frac{R_j^{(n)}}{R_j} \right].$$

(13)

To infer the variance of the negative binomial distribution, we can also leverage our neutral barcodes. We can think about all the potential sources of noise in the stages from cell number in the flask to read count. There is biological growth noise

and bottlenecking noise, which should be correlated across timepoints, and there is uncorrelated technical noise from DNA extraction, 30 rounds of PCR, and sequencing. All of these processes are counting processes, so we expect the variance of the frequencies to be proportional to the mean. Note, we are not treating the total coverage at a particular time point as a random variable. So we have

$$\mathrm{var}(f_{i,t}) \propto \frac{\langle r_{i,t} \rangle}{R_t} = c_t \frac{\langle r_{i,t} \rangle}{R_t}. \tag{14}$$

Given that we have a variance that is strongly dependent on the mean, we apply a variance-stabilizing transformation. Let's define

$$\phi_{i,t} \equiv \sqrt{f_{i,t}}. \tag{15}$$

Applying the square root transformation effectively decouples the variance from the mean. To see this, we can think about how the variance in $f_{i,t}$ is related to the variance in $\phi_{i,t}$. We can start with the Langevin equation for the change in frequency of a mutation due to drift.

$$f(t + \delta t) = f(t) + \sqrt{\frac{f(t)(1 - f(t))\delta t}{N_e}} Z \tag{16}$$

Consider the change in $\phi$ over one time interval:

$$\begin{aligned}
\delta\phi &= \phi(t + \delta t) - \phi(t) \\
&= \sqrt{f(t + \delta t)} - \sqrt{f(t)} \\
&= \sqrt{f(t) + \sqrt{\frac{f(t)(1 - f(t))\delta t}{N_e}} Z} - \sqrt{f(t)} \\
&= \sqrt{f(t) + \sqrt{\frac{f(t)\delta t}{N_e}} Z} - \sqrt{f(t)} \\
&\quad (f \ll 1, \ \sqrt{\frac{f(t)\delta t}{N_e}} Z \ll 1) \\
&= \sqrt{f(t)} \left( 1 + \frac{1}{2}\sqrt{\frac{\delta t}{N_e}} Z \right) - \sqrt{f(t)} \\
&= \sqrt{\frac{f(t)\delta t}{4 N_e}} Z.
\end{aligned}$$

So the variance in $\phi$ due to genetic drift over a single cycle is $\frac{1}{4 N_e}$. Between two timepoints, we expect neutral lineages to change due only to noise from genetic drift (which accumulates over time) and technical noise at each time point (which is independent at each time point). If there are enough read counts that the central limit theorem applies, we can add the variances:

$$\kappa_{t,t'} = \mathrm{var}(\phi_{i,t} - \phi_{i,t'}) = \eta_t + \eta_{t'} + \frac{|t - t'|}{N_e}. \tag{17}$$

We can measure $\kappa$ for each time interval, and treat the $\eta_t$ and $N_e$ as unknown parameters that we can fit by minimizing the squared errors of the expected relationship. We can also do this with inverse variance weighting by bootstrapping over barcodes when estimating $\kappa$.

$$\vec{\eta}, N_e = \arg\min_{\hat{\eta},N_e} \sum_{t,t'} \frac{\left(\eta_t + \eta_{t'} + \frac{|t-t'|}{4N_e} - \hat{\kappa}_{t,t'}\right)}{\text{var}(\hat{\kappa}_{t,t'})}. \tag{18}$$

We only use timepoints and barcodes with read count exceeding $r_{i,t} = 50$.

We can relate these quantities back to the variance in frequency, and obtain an estimate for the dispersion parameter, $c_t$, where

$$c_t = (4\eta_t + \frac{1}{N_e})R_t. \tag{19}$$

Having obtained our parameter estimates for intermediate parameters, we can finally infer the parameter value for fitness, $s$, using the likelihood function for a negative binomial random variable. The likelihood for a particular value of $s$ and $f_{0,i}$ given a barcode read count of $r_i$, is

$$P(r_i|s, f_{0,i}) = \prod_t \frac{\Gamma(r_{t,i} + \frac{\mu_{t,i}}{c_t-1})}{\Gamma(r_{t,i} + 1)\Gamma(\frac{\mu_{t,i}}{c_t-1})} \frac{(c_t - 1)^{r_{t,i}}}{c_t^{r_{t,i}+\frac{\mu_{t,i}}{c_t-1}}}. \tag{20}$$

We can marginalize over values of $f_0, i$

$$\mathcal{L}(s|r) = \prod_i \int df_{0,i} P(r_i|s, f_{0,i}), \tag{21}$$

and find the maximum likelihood fitness,

$$\hat{s} = \arg\max_s \log \mathcal{L}(s|r). \tag{22}$$

We use a grid search to find the value of $s$. We can use the likelihood function to infer a standard error on our parameter estimation, approximated as the inverse of the square-root Fisher information,

$$\text{std } \hat{s} = \frac{1}{\sqrt{-\partial_s^2 \log \mathcal{L}(s|r)|_{\hat{s}}}}. \tag{23}$$

To get a fitness estimate for a single environment, we infer fitness separately for each replicate, and then take a weighted average across our two biological replicates, weighted by the inverse of the inferred standard error.

**4.2.2 Fitness processing.** There were some technical problems in sequencing, resulting in several environments with poor data quality. We dropped these from further analysis. These included all temperature perturbations to the 2 day base (all Batch 1 conditions), the Batch 4 Salt Base, and 10 $\mu$M H89 on the Salt base.

**4.2.3 PCA on environment space.** To assess environmental similarity, we did principal component analysis on our fitness data. First, we took fitness data for all environments and centered and scaled it using

`sklearn.preprocessing.StandardScaler`. We then used `sklearn.decomposition.PCA` on the transpose of our standard fitness matrix (which normally has columns as environments and rows as mutants) so that we could understand environmental variation. We projected each environment onto the first two PCs and used `scipy.spatial.ConvexHull` to generate the convex hulls around each base environment.

**4.2.4 Calculation of z-score.** To quantify the magnitude of our perturbations, we used batch replicates as a baseline for variation. Each batch of our fitness measurements included one "unperturbed" base. For each base $B$, we took the collection of batch replicates and calculated the mean fitness for each genotype ($\vec{g}$), $\mu_B(\vec{g})$. We obtained an observed standard deviation ($\sigma_o(\vec{g})$) from these replicates, and we used the average inferred standard deviation from the fitness inference for the replicates ($\hat{\sigma}(\vec{g})$), together to represent variation across batch replicates. For each genotype $\vec{g}$ in each environment $\vec{E}$ belonging to base $B$, we calculated a z-score as

$$z_{\vec{E}}(\vec{g}) = \frac{|X(\vec{g}, \vec{E}_B + \delta\vec{E}_p) - \mu_B(\vec{g})|}{\sqrt{(\sigma_o^2(\vec{g}) + \hat{\sigma}^2(\vec{g}))}}.$$

(24)

To characterize the entire environment, we took the average $z_{\vec{E}_B + \delta\vec{E}_p}(\vec{g})$ across all mutants, and the standard deviation across all mutants. For the external bases, we used the same equation for $z_{\vec{E}}(\vec{g})$ as above, but inserted environments corresponding to different bases.

**4.2.5 Generating the $\delta X$ matrix.** To generate the $\delta\mathbf{X}$ matrix, we use the numerator of Eq. (24), without taking the absolute value. Thus, for each perturbation $p$, we define

$$\delta X_p^B(\vec{g}) = X(\vec{g}, \vec{E}_B + \delta\vec{E}_p) - \mu_B(\vec{g}).$$

(25)

We propagate the error in each measurement (both in generating the standard error of the base environment mean, and in the subtraction of fitness values to generate $\delta X_p^B(\vec{g})$.

**4.2.6 Calculating deviation from 1:1 line of perturbations on different base environments.** For each mutant, we compared the change in fitness due to the same perturbation, on different base environments (as shown in Fig 3B). We calculated a deviation from the 1:1 line by subtracting one base's $\delta X_p$ from another.

$$\delta X_p^{B_1}(\vec{g}) - \delta X_p^{B_2}(\vec{g}) = X(\vec{g}, \vec{E}_{B_1} + \delta\vec{E}_p) - \mu_{B_1}(\vec{g}) - X(\vec{g}, \vec{E}_{B_2} + \delta\vec{E}_p) - \mu_{B_2}(\vec{g}).$$

(26)

Each panel of Fig 3A compares $\delta X_p^{1\,\text{Day}}$ and $\delta X_p^{2\,\text{Day}}$, $\delta X_p^{1\,\text{Day}}$ and $\delta X_p^{\text{Salt}}$, and $\delta X_p^{2\,\text{Day}}$ and $\delta X_p^{\text{Salt}}$, respectively. We also calculated

$$\delta X_R = \delta X_{\text{Rep 1}}(\vec{g}, \vec{E}) - \delta X_{\text{Rep 2}}(\vec{g}, \vec{E})$$

(27)

$$= X_{\text{Rep 1}}(\vec{g}, (\vec{E}_B + \delta\vec{E}_p)) - \mu_B(\vec{g}) - (X_{\text{Rep 2}}(\vec{g}, (\vec{E}_B + \delta\vec{E}_p)) - \mu_B(\vec{g})).$$

(28)

This last comparison was to have a point of reference for the deviation to be expected under biological and measurement noise, and not due to ExE interactions. For each base-base comparison, we only included replicate-replicate $\delta X_p$ values belonging to the two bases in question, to prevent spurious inference of signal from noise.

**4.2.7 Quantifying deviations from 1:1 line.** For the single mutant case, we wanted to compare individual perturbations on the $\delta X$ scatter plots to the cloud of replicate-replicate points. To do so, we calculated the Mahalanobis distance, which is essentially a multivariate z-score. The Mahalanobis distance is

$$D(\vec{x}, Q) = \sqrt{(\vec{x} - \vec{\mu})\Sigma^{-1}(\vec{x} - \vec{\mu})},$$

(29)

where $Q$ is a probability distribution with a mean $\vec{\mu}$ and a positive semi-definite covariance matrix $\sigma$. To determine which perturbations differed significantly from the replicate-replicate distribution, we leveraged the fact that the squared Mahalanobis distance follows a $\chi^2$ distribution in the null case, and calculated the p-value as

$$p - \text{value} = \mathbb{P}(\chi^2(k) \geq D^2) = 1 - \text{CDF}(D^2; k), \tag{30}$$

where the CDF is the cumulative distribution function of the $\chi^2$ distribution with $k$ degrees of freedom, evaluated at $D^2$. We said that a perturbation exhibited significant ExE effects if deviated significantly from the replicate-replicate distribution on any of the base environment comparisons. We allow for batch replicates of the base environment to count as "perturbations" from the mean across base environments because batch-to-batch variation can arise from true biological variation [51].

We used the two sample Kolmogorov–Smirnov statistical test to determine whether the distribution of deviations from the 1:1 line was significantly different between the pairwise base comparisons and the replicate-replicate comparisons for the aggregation of all mutants. We used `scipy.stats.ks_2samp` to obtain the value for the statistic. To assess significance, we used a randomized bootstrapping approach, in which we resampled from both distributions with replacement, calculated the KS distance, and counted how many times the randomized statistic was equal to or larger than the observed statistic. Additionally, we used the 5th and 95th quantiles to determine how many base-base comparisons were significantly different from the distribution of replicate-replicate comparisons, with a 10% false discovery rate (FDR).

**4.2.8 Singular value decomposition on $\delta X$ matrices.** To generate our model, we use singular value decomposition to factorize the $\delta X$ matrix. For each base, we perform SVD using `numpy.linalg.svd`. We determine fraction of variance explained by component in Fig 4 by taking each singular value squared, and dividing by the sum of squared singular values:

$$\text{Frac. var. explained by } i = \frac{\sigma_i^2}{\sum_j \sigma_j^2}. \tag{31}$$

- First, for focal scree plot, we take only the set of adaptive mutants from Kinsler and colleagues [28]. Then, we set a standard error threshold (here, we used 0.3), where for a particular mutant, if it has an inferred error on its fitness greater than the threshold, we exclude it. This is to avoid noise causing a spike in signal that will corrupt the dimensionality plots. This is of course subject to our choice of threshold, but for a variety of thresholds we find similar numbers of mutants to exclude. In this case, we exclude 29 mutants out of 258.

- We next take each set of environments, remove No Shake from Salt because we saw that it was not subtle, and do SVD on the resulting matrix

- We obtain singular values and calculate fraction of variance explained for each singular value.

**4.2.9 Calculating a limit of detection for dimensionality.** To calculate the number of dimensions that exceed noise in our data, we take the same approach as [28]. We use the inferred standard error in our fitness estimate, and generate 1,000 random matrices of the same shape as our data, centered at 0. Each entry is drawn from a normal distribution with mean 0 and standard deviation equal to the inferred standard error at the corresponding index in our data. From this, we can ask how much variation in the true data would have been explained by components derived from the noise-only matrix, by taking the singular values squared from the noise matrix and dividing by the sum of squared singular values in the data. The maximum amount of variation explained by noise matrices represents our limit of detection:

$$\text{Frac. of var. explained by noise} = \frac{\bar{\sigma}_{1n}^2}{\sum_j \sigma_j^2}. \tag{32}$$

where $\bar{\sigma}_{1n}$ is the average first singular value across all folds of the noise-only matrices, and the $\sigma_j$ in the denominator come from the $\delta X$ matrix.

**4.2.10 Linear regression on fitnotypes.** On a methodological note, it might seem unnecessary to approach overlap between vectors via prediction, rather than directly looking at some quantity like the inner product, or cosine of the angle between the two $\vec{\phi}$ vectors. We take this approach because the method we use to infer our $\vec{\phi}$ vectors is matrix factorization, which is under-determined. This means that any rotation or linear transformation of our latent fitnotype space will be equally good at reconstructing our data, so we must focus on properties of our model that are invariant under linear transformation. Prediction error is one such quantity. See section S1 for more detail on this point.

We use prediction error for a linear model trained in one base, and tested in held out environments, to asses model success and overlap in latent space.

**Set-up and leave-one-out SVD** First, we do SVD on $\delta X$ all environments in one base, but we leave out a single perturbation. Let $B_1$ be our focal base, and $B_2$ is a target base. For a perturbation $p$, we have the reduced matrix $\delta X^{B_1}_{-p}$, where we have removed perturbation p. We then do SVD on this reduced matrix. Here,

$$\delta \mathbf{X}^B \in \mathbb{R}^{n_{\text{mut}} \times n_{\text{pert}}}, \qquad \delta X^B_p := \delta \mathbf{X}^B[:, p] \in \mathbb{R}^{n_{\text{mut}}}. \tag{33}$$

For the focal base $B_1$ and held-out perturbation $p$,

$$\delta X^{B_1}_{-p} = \mathbf{U_{-p}} \Sigma_{-\mathbf{p}} \mathbf{V}^{\mathsf{T}}_{-\mathbf{p}}. \tag{34}$$

For each value of $k$ up to $k = 6$, we use the first $k$ columns of $\mathbf{U_{-p}}$, $\vec{u}_{:k}$, as features in a linear regression, fitting coefficients for the held out perturbation $p$ within base:

$$\text{training base loss } \mathcal{L}_k(\text{train}) = ||\vec{u}_{:k,-p} \cdot \hat{\beta}_{k,B_1} - \delta X^{B_1}_p||^2. \tag{35}$$

We calculate the variance explained by component by looking at the squared sum of errors (SSE) and the total sum of squares (TSS), and get variance explained by the $k$ dimensional model as:

$$\text{fraction of variance explained for } k\text{-dim model} = 1 - \frac{SSE_k}{TSS}. \tag{36}$$

We can do almost the same procedure for the target base. This time, though, we use the full focal base for the SVD, $\delta X^{B_1} = \mathbf{U}\Sigma\mathbf{V}^{\mathsf{T}}$. Here,

$$\text{target base loss } \mathcal{L}_k(\text{target}) = ||\vec{u}_{:k} \cdot \hat{\beta}_{k,B_2} - \delta X^{B_2}_p||^2. \tag{37}$$

We calculate the fraction of variance explained in the target base perturbation in the same way. We can then compare predictive power across components and across bases, as in Fig 5B and C. Note, within base, we are actually training on slightly different models for each held out prediction, and aggregating predictive power across. By comparing the unexplained variation across bases, and the weights of each component, we can glean information about the underlying fitnotype spaces, even though they are latent. In doing this procedure, we assess how well the $B_1$ subspace spans the $B_2$ columns after refitting coefficients on $B_2$. This measures subspace overlap, independent of coefficients.

**4.2.11 A null model for predictive power of linear regression on fitnotypes..** We wish to estimate how these linear regressions look when the data are purely generated from noisy processes, rather than meaningful biological variation. We take a similar approach to the null we use in Fig 3. We assume the difference in $\delta X$ between replicates should be

drive purely by noise. These are biological replicates, not technical replicates, so it is theoretically possible that there is meaningful biological variation between the two flasks, but the conservative choice is to assume that differences in $\delta X$ between replicates are pure noise. We calculate $\delta X_{\text{Rep 1}} - \delta X_{\text{Rep 2}}$ for all base environments and all perturbations. We use that output as the input for the procedure described in section 5.2.10.

## Supporting information

**S1 File. Supplementary information text.** File containing sections expanding upon analyses presented in the main text.
(PDF)

**S1 Fig. Table of environmental perturbation components.** Here, we present a detailed description of each environment and perturbation. We color the rows by base environment, and specify all variants of the environment, along with which batch of the experiment included each perturbation.
(PDF)

**S2 Fig. ExE interactions across all mutants. (a)** 2-dimensional histograms of due to focal perturbation on two different base environments. The color of pixel corresponds to the number of mutants in the bin. The top right shows correlation coefficient for each environmental comparison. **(b)** Correlation matrix between environmental perturbations, clustered by perturbation (then batch) (top) and base environment (bottom). Block diagonal form is more apparent on the bottom, suggesting that the base environment is important for determining $\delta X$. The data underlying this figure can be found in https://zenodo.org/records/18175930.
(PDF)

**S3 Fig. Scree plots for second step mutants.** Each set of mutants has a different ancestor, and a different evolution condition. We did SVD on each set of mutants in each base environment, and here show the fraction of variance explained by each component for each base. Green is 2 Day, blue is 1 Day, and pink is Salt. The data underlying this figure can be found in https://zenodo.org/records/18175930.
(PNG)

**S4 Fig. Quantifying dimensionality in alternate ways.** Inferred dimensionality in evolution environment base plotted versus dimensionality inferred in alternate base. **(a)** Dimensionality is inferred based on how many components fall above the most explanatory noise-only matrix for all the bases. **(b)** Entropy of the distribution of variance explained for the components that fall above the overall limit of detection across bases is used as a proxy for dimensionality. **(c)** Entropy of distribution of variance explained for components that explain more than each base's individual detection limit is used as a proxy for dimensionality. The data underlying this figure can be found in https://zenodo.org/records/18175930.
(PDF)

**S5 Fig. Titrating in perturbations and inferring dimensionality.** Each panel is a distinct set of mutants, evolved from a different ancestor and in a different environment. Across a range of $n$, we still find that the evolution condition is not systematically lower in inferred dimensionality than the other two base environments. X and Y jitter added. The data underlying this figure can be found in https://zenodo.org/records/18175930.
(PDF)

**S6 Fig. Synthetic data. (a)** Underlying weights of fitnotypes (rows) in different environments (columns) for different scenarios of fitnotype overlap. **(b)** Fitness matrices for the same mutant-to-fitnotype matrix, but different environment-to-fitnotype matrices (from panel a). **(c)** Variance explained by each inferred fitnotype using SVD to identify fitnotypes. **(d)** Prediction within and across bases for different fitntoype overlap. The data and code underlying this figure can be found in https://zenodo.org/records/18175930.
(PDF)

**S7 Fig. Predicting** $\delta X$ **with linear regression for different mutant sets.** Columns correspond to predictions for different mutant sets (mutants that evolved from different ancestors). We show results for training base 2 Day **(a)**, training base 1 Day **(b)**, and training base Salt **(c)**. The data underlying this figure can be found in https://zenodo.org/records/18175930. (PDF)

**S8 Fig. Predictions for target bases separated out by perturbation.** Each prediction is done for one perturbation. The contribution of each component from the training base to predicting $\delta X$ in the test perturbation is shown here. **(a)** Training base is 2 days. **(b)** Training base is 1 day. **(c)** Training base is Salt. The data underlying this figure can be found in https://zenodo.org/records/18175930. (PDF)

**S9 Fig. Model Comparison. (a)** Comparison of model predictions and measured $\delta X$ for 2 day adaptive mutants, using a gene-only model, a perturbation-only model, and a linear fitnotype model (BCV), for two environments. **(b)** $R^2$ for all environments for each model, colored by base environment. The $\delta X$ is least predictive on average, and the BCV is most predictive on average, despite heterogeneity. The data underlying this figure can be found in https://zenodo.org/records/18175930. (PDF)

**S10 Fig. Alternative geometric schematic for our linear regression-based approach for comparing fitnotypes.** (TIFF)

**S11 Fig. Here, we perform the same analysis as in Fig 5 in the main text, but for the noise matrix described in Methods section 5.2.11.** We use linear regression to predict fitness on a purely noise-derived matrix. The difference in $\delta X$ between replicates is the "fitness" value, in order to generate a null for how much predictive power we expect by chance. The data underlying this figure can be found in https://zenodo.org/records/18175930. (PDF)

## Acknowledgments

We would like to thank members of the Petrov lab and Good lab for helpful discussions and feedback. We would like to thank M. Tikhonov, G. Mel de Fontenay, A. Thomas, M. Aguirre, J. Pritchard, S. Kuehn, S. Kryazhimskiy, D. Stanford, M. Johnson, D. Wong, A. Khristich, and D. Fisher for helpful discussions. We would like to thank M. Tikhonov, C. Abreu, S. Mathur, A. Khristich, A. Lyulina, V. Chen, K. Aguilar-Rodriguez, S. Walton, M. Razo-Mejia, and Z. Liu for their help with the experiments (Big Batch Bootcamp team). We would like to thank R. Stelkens, K. Geiler-Samerotte, H. Fraser, M. Siegal, and C. Caniglia for feedback on the manuscript. We thank the Stanford Research Computing Center for use of computational resources on the Sherlock cluster.

## Author contributions

**Conceptualization:** Grant Kinsler, Dmitri A. Petrov.

**Data curation:** Olivia Ghosh, Grant Kinsler.

**Formal analysis:** Olivia Ghosh, Grant Kinsler, Benjamin H. Good.

**Funding acquisition:** Dmitri A. Petrov.

**Investigation:** Olivia Ghosh, Grant Kinsler.

**Methodology:** Olivia Ghosh, Grant Kinsler.

**Project administration:** Dmitri A. Petrov.

**Resources:** Dmitri A. Petrov.

**Software:** Olivia Ghosh, Grant Kinsler.

**Supervision:** Grant Kinsler, Benjamin H. Good, Dmitri A. Petrov.

**Validation:** Olivia Ghosh.

**Visualization:** Olivia Ghosh.

**Writing – original draft:** Olivia Ghosh.

**Writing – review & editing:** Olivia Ghosh, Grant Kinsler, Benjamin H. Good, Dmitri A. Petrov.

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
