## [Editor Report · Decision Letter 0]

3 Jul 2025

Dear Dr Ghosh,

Thank you for submitting your manuscript entitled "Low-dimensional genotype-fitness mapping across divergent environments suggests a limiting functions model of fitness" for consideration as a Research Article by PLOS Biology.

Your manuscript has now been evaluated by the PLOS Biology editorial staff, and I am writing to let you know that we would like to send your submission out for external peer review. I should say, however, that I was not able to secure advice from an Academic Editor in timely fashion, so we'll be looking for some enthusiasm from the reviewers.

Once your full submission is complete, your paper will undergo a series of checks in preparation for peer review. After your manuscript has passed the checks it will be sent out for review. To provide the metadata for your submission, please Login to Editorial Manager (https://www.editorialmanager.com/pbiology) within two working days, i.e. by Jul 07 2025 11:59PM.

Kind regards,

Roli Roberts

Roland Roberts, PhD

Senior Editor

PLOS Biology

rroberts@plos.org

---

## [Decision Letter · Decision Letter 1]

25 Aug 2025

Dear Dr Ghosh,

Thank you for your patience while your manuscript "Low-dimensional genotype-fitness mapping across divergent environments suggests a limiting functions model of fitness" went through peer-review at PLOS Biology. Your manuscript has now been evaluated by the PLOS Biology editors, an Academic Editor with relevant expertise, and by three independent reviewers.

You'll see that reviewer #1 is very positive, simply asking you to adjust a Figure. Reviewer #2 is also positive, but wants you to make the paper more streamlined and accessible, better connected to the existing literature, and with more clarity about assumptions. Reviewer #3 is positive, but has a long list of requests, most of which relate to issues of clarity, and some of which may involve additional analyses.

IMPORTANT: There are repeated mentions of lack of clarity, and I'm aware that this a somewhat abstract and conceptually challenging paper, but it is essential that you strive to convey your important ideas to our broad readership; hopefully the reviewers' comments will guide you in this direction.

In light of the reviews, which you will find at the end of this email, we are pleased to offer you the opportunity to address the comments from the reviewers in a revision that we anticipate should not take you very long. We will then assess your revised manuscript and your response to the reviewers' comments with our Academic Editor aiming to avoid further rounds of peer-review, although we might need to consult with the reviewers, depending on the nature of the revisions.

**IMPORTANT - SUBMITTING YOUR REVISION**

*Resubmission Checklist*

*Published Peer Review*

*PLOS Data Policy*

Sincerely,

Roli Roberts

Roland Roberts, PhD

Senior Editor

PLOS Biology

rroberts@plos.org

REVIEWERS' COMMENTS:

Reviewer #1:

This paper takes on a rather broad set of questions: can we infer how many "fundamental" phenotypes contribute to fitness in a given environment and how does this change with respect to adaptation? They take a very statistical approach to this quite abstract (from the point of view of many biologists) question and do it about as much justice as I think is possible in the writing. They really try to make the concepts accessible and largely succeed.

The premise of the paper is a very original experimental design of three "home environments" and a panel of 20 perturbations from each. Then they throw the same panel of barcoded mutant strains at them and examine the data. As they come back to in the discussion, the three home environments are not THAT different from each other, but that is a common challenge in these types of experiments, whereby we tend to hold almost all variables constant and do many perturbations of just one or two things (like substrate and stressor while keeping EVERYTHING else the same). Despite this, I think the approach and the results are pretty compelling. They soundly reject the idea that the selective environment is much less multi-dimensional than alternative ones in the pleiotropic effects that arise. In fact, the data turn out to generally be the opposite.

Whether we can predict from mutations to fitness remains unclear, but this does raise hope that we can predict fitness in one environment given fitness in another (as they tackle in Figure 5). This suggests that there are a reasonably finite number of currencies/processes that all of the details in the cell trickle down into.

This may not be a paper that many molecular biologists can conceptualize well, but I think it is very well done and is quite original. We need more papers like this that stretch one's brain to abstract what it is that we do. The biggest question is probably whether people "buy it" that regression results correlate with actual definable biological processes, but that is a goal well beyond this current work.

Minor comment:

Figure 2 - I think this heat map is very hard to interpret. I want to be able to assess whether mutants are better or worse than the WT ancestor, yet it is a one-color scale. Neutrality is thus a particular color of blue and it is hard to assess all of the mid-tones for being higher or lower than that (the extremes are easy). Please change to a two color scheme with white in the middle and shades of red and blue (or whatever) to indicate higher or lower than WT. I think that will end up being quite striking.

Reviewer #2:

I have read the manuscript carefully and assessed it to the best of my ability. While I am familiar with the general concepts and biological context, my expertise does not fully cover all of the specific analytical aspects employed here. My comments should therefore be considered in light of this limitation.

This manuscript presents an elegant experimental and analytical approach to investigating genotype-phenotype-fitness relationships in adaptive yeast mutants exposed to multiple environments. By combining singular value decomposition and regression analyses, the authors identify a low-dimensional structure in the fitness landscape and propose a "limiting functions" model of fitness, in contrast to the "pleiotropic expansion" hypothesis. The study is timely, methodologically sound, and addresses a central question in evolutionary biology concerning the constraints on adaptation across environments.

The work has several strengths. The experimental design is robust, with appropriate replication and carefully chosen environmental contrasts. The statistical and modelling approaches are well-justified, and the figures are generally clear and informative. Importantly, the proposed "limiting functions" model is novel and has broad implications for understanding the evolutionary trade-offs that shape adaptation.

There are, however, areas where the manuscript could be improved. The data analysis, while rigorous, is highly technical and may be difficult for a broad readership to follow; streamlining this section could increase accessibility. The discussion would benefit from deeper engagement with previous literature on trade-offs and pleiotropy, situating the findings more explicitly in the context of long-standing theoretical frameworks. Finally, certain assumptions underlying the regression and singular value decomposition analyses should be stated and justified explicitly.

Reviewer #3:

Overall: The manuscript from Ghosh et al demonstrates a interesting idea that a "pleiotropic shift" picture of how mutations affect fitness in one environment + perturbations maps onto the effect of those mutations on fitness in another base environment + perturbations. The study comes with a massively parallelized barcode based measurement of fitness on mutants is 10s of environments. The dataset is substantial, the analysis of fitness is quantitative, as is the analysis of high level patterns in the dataset. I think the study warrants publication. My comments below reflect aspects of the work that I think could be improved both conceptually and technically.

Major

1. In title and abstract -- "functions" means something like processes inside the cell? The title is especially confusing in this regard on first read. I think the authors should think a bit harder about how to write a title that is immediately accessible to a broader audience.

a. Similarly -- what concretely are "nonlimiting functions"?

b. What do we mean by "local" and "global" in the abstract? Is this local with respect to the environment? The mutation?

c. The abstract is genetics jargon heavy, which leaves a wide readership out in the cold. I suggest removing the jargon and spelling out the key results in accessible language.

2. The introduction dives quickly into technical details in paragraph 2, and too much jargon in my opinion. For a broader audience it might be worth discussing the observations over the past several decades of low-dimensional structure at various levels of biological organization -- from proteins, to genomes, to physiology. It feels like I am reading an addendum to a previous study by this group rather than reading a stand-alone study.

3. Other comments on the introduction/framing.

a. Prior to equation 1 - would appreciate line numbers - "Despite this challenge, recent studies have hinted that there may be low-dimensional structure in genotype-phenotype maps (16-23). In most of these studies, the fitness of a particular genotype is portrayed as linear function of K underlying, latent phenotypes" -- is confusing because the claim of low-D genotype to phenotype maps is followed by an equation mapping latent phenotypes to fitness, not genotypes to phenotypes.

b. One seemingly important point left out is the linearity of eq. 1 -- the discussion revolves around this linear mapping. Perhaps there is a low-dimensional but non-linear relationship that looks more complex due to the assumptions of the SVD used to infer these fitnotypes.

c. The framework places these 'fitnotypes' on a different footing entirely from phenotypes. It is unclear to me if this assumption is justified. These fitnotypes could be phenotypes, but we simply do not know which phenotypes due to the construction of the experiment/model. Do you agree?

d. In the pleiotropic shift/expansion -- presumes that there are many phenotypes involved in determining fitness in the experiment. This is an untested assumption which I think has little definitive empirical evidence. One can of course argue that many subcellular biological processes might matter for fitness, but this qualitative picture isn't something that has been definitively shown, and therefore I think it needs to be stated as an assumption and not as a given. The reason this is important is that the pleiotropic expansion model assumes exactly this, that there are many latent "knobs" the cell can turn. I am saying that these models which make assumptions about the dimensionality of phenotypes might be relatively weak straw men.

e. In part what I find so confusing about the pleiotropic expansion model is that it would also seem to imply another model (not considered) which is a pleiotropic "compression" -- e.g. going from one base to another actually reduces dimensionality (depending on the base you choose to start with). Wouldn't you expect this to happen in some cases where the base environment has a high dimensional latent space and the "distant" environment a low D space? Why would one only expect expansion? This is in fact partially what is observed in Figure 4.

4. Equation 2 -- it would be useful to define your notation in equation 2 better. The subscript p is for perturbation? (but this is different from equation 1 where the index i is for mutations) what is the average in \bar{X} in the last term exactly? (Digging into the SI, it looks like it might be worth defining the \mu from Eqn. 24 in the main text to avoid confusion). The other thing that is not indexed is replicates. This makes the paragraph starting "technical noise.." hard to follow and my understanding of Fig. 3a perhaps incorrect.

a. The interpretation of Figure 3a and b is challenging. The authors are pointing out conditions that deviate strongly from the 1:1 line, however almost every point differs from the 1:1 significantly if I am using the plotted error bars to eyeball significance.

b. Second, the paragraph in question "Technical noise.." says that you are plotting "replicate to replicate" correlations….but I interpret the x-axis of the upper left panel in Fig. 3a as "Pert. Fit. Effect 1 Day base." -- so it would seem you are plotting not replicates but fitness in 1 day + perturbation vs 2 day + perturbation (delta defined via eqn. 2). In that sense, how can the spread around this 1:1 line in this panel be interpreted as noise? In SI section 5.2.6 it sounds like you would like to plot delta X_p^Rep1 vs delta X_p^Rep2 -- but is that what is shown in this panel? This is confusing, because when I try to understand panel (b) the gray distribution is Rep1 - Rep2 with a spread of about 0.5 units…..where can I see the scatterplot for a given mutation of Rep 1 vs Rep 2?

5. "We next aggregated the deviation from additivity (residuals from 1:1 line) for all mutants in all environments and compared them to this null expectation." I do not follow. 1:1 line doesn't define additivity, it defines equal delta X for a given mutation in different environments. Second, what null?

a. Seems like the spread of figure 3b should somehow be shown in the panels in figure 3a, because my comment 3a pertains -- it seems like error bars are ~0.1, but spread in gray distributions in (b) are of order 0.5.

6. In the "next we aggregated" paragraph-- the KS is returning the same astronomically low p-value for all comparisons. This test is super sensitive to small changes in distributions -- what we care about is actually which comparisons deviate from the Rep1 - Rep2 null. I would make a comparison using a bootstrapping randomization scheme rather than this non-parametric test.

7. Section 3.4 -- I find the introduction of the additional mutants to be out of place in this section. The statement is that in each base environment the dimensionality of the delta X's is relatively low. Putting these mutants in at the top of p14 is distracting. Would suggest removing.

8. Section 3.5 --

a. The prediction framework as a test for the model proposed in 1 - is it the right choice? My read of Figure 1c.ii is that one wants to ask whether the latent variables are *the same* or different for a distant environment. In essence, are the \phi_ikB vectors the same when inferred on two different environments? Since the same mutants are used across all base environments, can you not make a comparison between these vectors are bases? Dot products relative to some null could tell you if these vectors are "aligned" or not. This seems important, because in the regression formalism you perform (as I understand it, section 5.2.10) -- you have an added degree of freedom which are the coefficients in the linear regression models -- this regression approach answers the question "can the latent phenotypes in one base predict a held out perturbation?" but not "are the latent variables similar across bases". In this sense, can I really interpret the outcome of these regressions as is shown schematically in figure 5b i - iii? E.g. if predictions are bad are the latent variables orthogonal? In this sense it isn't clear to me if we answered the question of whether or not the latent phenotypes are the same or different in two environments.

b. What are the "green" fitnotypes?

c. Can you construct a null for these regressions that uses the replicates as in Figure 3? I want to understand how much of the variance I would expect to be included in the diagonal hashed regions. Is what we are seeing here a lot or a little? Can you construct such a null?

d. Figure 5 is a masterclass in data presentation. Well done!

9. Discussion --

a. Two paragraphs top half of p 19 -- brings up an important point which I think is not sufficiently discussed in the paper. Perhaps one explanation is that there really aren't that many phenotypes that can be modified to affect fitness in this experiment. Perhaps what you are seeing is that a few physiological properties of the cell (lag, growth rate, entry into stationary phase, core stress response) determine growth in all of your experiments and these latent variables are reflections of this essential low dimensionality. This is similar to comment 3.d.

10. Physiological comment on dimensionality -- one might speculate that the reason for the low dimensionality inferred in the the delta X in this experiment is because of the limitations in the perturbations chosen. If I look at the list of perturbations, I see that they are quite restrictive -- for example, all carbon substrates are sugars (except ethanol), varying the quantity of glucose is a perturbation simply in the carrying capacity (and duration in stationary phase), similarly not shaking is limiting oxygen availability, and then the drugs are stresses. So perhaps the low dimensionality comes from the limited small perturbations that are applied -- e.g. carbon sources are glycolytic, oxygen limitation vs not affects respiration, stationary phase duration is varied and then the drugs. Is it not plausible that the results come from this limitation in the experiment? I realize doing many perturbations is itself a hard experiment, I'm not proposing this be done, I'm suggesting that this possibility perhaps warrants a discussion.

a. Related, there is a hesitance to discuss the actual phenotypes governing fitness in the experiment (related to 3.c). Given the limited scope of perturbations, I am left wondering if the phenotypes that these mutations are targeting aren't a bit simpler than the authors make it sound. Perhaps these mutations target a few key phenotypes that matter in these environments (resp vs fermentive metabolism, lag, stress survival) -- and these might actually be learnable from a targeted set of experiments on mutants in given environments -- indeed, if the 'limiting functions' that govern fitness are in spirit similar to those limiting functions discussed (e.g. Leibig) then these may well be discoverable. This is at odds with the language throughout the manuscript that postulates the existence of a very high dimensional phenotype space with many phenotypes potentially impacting fitness (e.g. via the expansion hypothesis). Perhaps this is one hopeful outcome of this study, that these limiting functions are themselves understandable in terms of concrete traits.

Minor

1. The title to Fig. 3a is also confusing. "Mutant 1: WT ancestor; IRA1 mutation," what does it mean exactly? Just the IRA1 mutation in the WT background? On first read it sounds like one mutation is the WT ancestor and another mutation is IRA1, but this cannot be true because we are looking at fitness relative to the ancestor for the IRA1 mutation. Please clarify this title.

2. Figure 3 caption "would predict indicate" typo

3. In figure 3b is there some sort of kernel density estimate or smoothing being done to the data? If so it should be mentioned in the caption.

4. The null model for SVD constructed at the bottom of p 13 -- why use Gaussian random variables and not just the data itself, destroying any correlation? The reason for this question is that this null brings up the question of whether the data are Gaussian or not….

---

## [Decision Letter · Decision Letter 2]

11 Dec 2025

Dear Dr Ghosh,

Thank you for your patience while we considered your revised manuscript "Genotype-fitness mapping of adaptive mutants reveals shifting low-dimensional structure across divergent environments" for publication as a Research Article at PLOS Biology. This revised version of your manuscript has been evaluated by the PLOS Biology editors, the Academic Editor, and one of the original reviewers.

Based on the review and on our Academic Editor's assessment of your revision, we are likely to accept this manuscript for publication, provided you satisfactorily address the following data and other policy-related requests.

IMPORTANT - please attend to the following:

a) This may just be me, but I couldn't spot a citation to Fig S10; please check and rectify if necessary.

b) Please address my Data Policy requests below; specifically, we need you to supply the numerical values underlying Figs 2ABCD, 3AB, 4AC, 5C, S2ABC, S3, S4ABC, S5, S6ABCD, S7ABC, S8ABC, S9AB, S11ABC, either as a supplementary data file or as a permanent DOI’d deposition. I note that you already have an associated GitHub deposition (https://github.com/omghosh/limiting-functions). Please could you confirm whether the data and code in this deposition are sufficient to recreate the Figures? Also, because Github depositions can be readily changed or deleted, please make a permanent DOI’d copy (e.g. in Zenodo) and provide this URL (see below).

c) Please cite the location of the data clearly in all relevant main and supplementary Figure legends, e.g. “The data underlying this Figure can be found in https://zenodo.org/records/XXXXXXXX

d) Please make any custom code available, either as a supplementary file or as part of your data deposition. It looks like this is already in your Github (and so will be in your Zenodo deposition), but can you confirm?

e) Please include the URLs of your funders in the Financial Disclosure statement.

We expect to receive your revised manuscript within two weeks (do let me know if you need more time over the holiday season!).

*Published Peer Review History*

*Press*

Sincerely,

Roli Roberts

Roland Roberts, PhD

Senior Editor

rroberts@plos.org

PLOS Biology

DATA POLICY:

Regardless of the method selected, please ensure that you provide the individual numerical values that underlie the summary data displayed in the following figure panels as they are essential for readers to assess your analysis and to reproduce it: Figs 2ABCD, 3AB, 4AC, 5C, S2ABC, S3, S4ABC, S5, S6ABCD, S7ABC, S8ABC, S9AB, S11ABC. NOTE: the numerical data provided should include all replicates AND the way in which the plotted mean and errors were derived (it should not present only the mean/average values).

CODE POLICY

DATA NOT SHOWN?

REVIEWER'S COMMENTS:

Reviewer #3:

I have read the response from the authors. I feel they have done a very good job carefully responding to all of my queries and I have no further requests. I believe the manuscript is ready for publication.

---

## [Editor Report · Decision Letter 3]

12 Jan 2026

Dear Olivia,

Thank you for the submission of your revised Research Article "Genotype-fitness mapping of adaptive mutants reveals shifting low-dimensional structure across divergent environments" for publication in PLOS Biology. On behalf of my colleagues and the Academic Editor, Arjan de Visser, I'm pleased to say that we can in principle accept your manuscript for publication, provided you address any remaining formatting and reporting issues. These will be detailed in an email you should receive within 2-3 business days from our colleagues in the journal operations team; no action is required from you until then. Please note that we will not be able to formally accept your manuscript and schedule it for publication until you have completed any requested changes.

IMPORTANT: I've asked my colleagues to include the following request alongside your own: "Many thanks for citing your Zenodo URL in the main Figure legends; please could you also cite it (using similar wording) in the supplementary Figure legends?"

Sincerely,

Roli

Senior Editor

PLOS Biology

rroberts@plos.org